# Brief Communication: Evaluation and inter-comparisons of Qinghai-Tibet Plateau permafrost maps based on a new inventory of field evidence

Bin Cao[1,2], Tingjun Zhang[1], Qingbai Wu[3], Yu Sheng[3], Lin Zhao[4], and Defu Zou[5]

[1]Key Laboratory of Western China's Environmental Systems (Ministry of Education), College of Earth and Environmental Sciences, Lanzhou University, Lanzhou 730000, China
[2]Department of Geography & Environmental Studies, Carleton University, Ottawa K1S 5B6, Canada
[3]State Key Laboratory of Frozen Soil Engineering, Cold and Arid Regions Environmental and Engineering Research Institute, Chinese Academy of Sciences, Lanzhou 730000, China
[4]School of Geographical Sciences, Nanjing University of Information Science and Technology, Nanjing 210044, China
[5]Cryosphere Research Station on the Qinghai-Tibet Plateau, State Key Laboratory of Cryospheric Science, Cold and Arid Regions Environmental and Engineering Research Institute, Chinese Academy of Sciences, Lanzhou 730000, China

**Correspondence:** Tingjun Zhang (tjzhang@lzu.edu.cn)

**Abstract.** Many maps have been produced to estimate permafrost distribution over the Qinghai-Tibet Plateau (QTP), but the errors and biases among them are poorly understood due to limited field evidence. Here we evaluate and inter-compare the results of six different QTP permafrost maps against a new inventory of permafrost presence or absence comprising 1475 field sites compiled from various sources. Based on the in-situ measurements, our evaluation results showed a wide range of map performance with the Cohen's kappa coefficient from 0.21 to 0.58 and overall accuracy between about 55–83%. The low agreement in areas near the boundary between permafrost and non-permafrost and in spatially highly variable landscapes highlights the need for improved mapping methods that consider more controlling factors at both medium-large and local scales.

## 1 Introduction

Permafrost is one of the major components of the cryosphere due to its large spatial extent. The Qinghai-Tibet Plateau (QTP), also known as the Third Pole, has the largest extent of permafrost in the low-middle latitudes. Permafrost over the QTP was reported to be sensitive to climate change mainly due to high ground temperature ($> -2$ °C) (Wu and Zhang, 2008), and its distribution has strong influences on hydrological processes (e.g., Cheng and Jin, 2013; Zhang et al., 2018), biogeochemical processes (e.g., Mu et al., 2017), and human systems (e.g., Wu et al., 2016).

Many approaches have been used to produce permafrost distribution and ground ice condition maps at different scales over the QTP (Ran et al., 2012). Typically, these maps classify frozen ground into permafrost and seasonally frozen ground, and information on the extent, such as the areal abundance, of permafrost is available for some of them (Ran et al., 2012). These maps significantly improved the understanding of permafrost distribution over the QTP. However, limited in-situ measurements and the different classification systems and compilation approaches used make it challenging to compare maps directly. With

the availability of high-resolution spatial data sets (e.g., surface air temperature and land surface temperature), several empirical and (semi-) physical models have been applied in permafrost distribution simulations at fine scales (e.g., Nan et al., 2013; Zhao et al., 2017; Zou et al., 2017; Wu et al., 2018). The QTP has also been included in hemispheric or global maps including the Circum-Arctic Map of Permafrost and Ground-ice Conditions produced by the International Permafrost Association (referenced as IPA map) (Brown, 1997), and the global permafrost zonation index (PZI) map (referenced as $\text{PZI}_{\text{global}}$ map) derived by Gruber (2012).

Despite the increasing efforts in mapping QTP permafrost, the maps have not been evaluated and inter-compared with the large amount of permafrost presence or absence evidence. These data have been collected since the 2000s, and represent a number of different field techniques including ground temperature measurements, soil pits, and geophysics. A new inventory of this field evidence provides an opportunity to improve the evaluation of the existing permafrost maps. This is an important step in describing the current body of knowledge on permafrost mapping performance as well as identifying any possible bias. It is also critical for identifying priorities when updating these maps in the future. Additionally, an improved evaluation is a useful guide to selecting a map to use for permafrost and related studies, such as, setting boundary conditions for eco-hydrological model simulations. Climate change and increasing infrastructure construction on permafrost add both environmental and engineering relevance to investigating permafrost distribution, and increase the importance of evaluating and comparing existing permafrost maps.

In this study, we aim to:

1. provide the first inventory of permafrost presence or absence evidence for the QTP; and

2. use the inventory to evaluate and inter-compare existing permafrost maps on the QTP.

## 2  Data and methods

### 2.1  Inventory of permafrost presence or absence evidence

Four methods were used to acquire evidence of permafrost presence or absence: borehole temperatures (BH), soil pits (SP), ground surface temperatures (GST), and ground-penetrating radar (GPR) surveys (Figure 1, Table1). In this study, we used the mean ground temperatures (MGT) measured from the boreholes, the depths of which vary from meters to about 20 m depending on the depth of zero annual amplitude and borehole depth, to identify permafrost presence or absence. At SP sites, the presence of ground ice was used to indicate permafrost presence. However, due to the prevalence of coarse soil, there are only 6 SP sites and the depths range from less than 1 m to about 2.5 m. Thermal offset, here defined as the mean annual temperature at the top of permafrost (TTOP) minus the mean annual ground surface temperature (MAGST) at a depth of 0.05 or 0.1 m, was used to estimate permafrost presence or absence for sites with only GST available. Although it is spatially variable depending on soil and temperature conditions, the magnitude of the thermal offset is small on the QTP compared with northern, high-latitude environments due to the prevalent coarse soil and low soil moisture content. The maximum thermal offset under natural conditions reported for the QTP is 0.79 °C (referenced as maximum thermal offset, $\text{TO}_{\text{max}}$) (Wu et al., 2002, 2010; Lin

et al., 2015). In this study, sites with $MAGST + TO_{max} \leqslant 0$ °C are considered to be permafrost sites. The reversed thermal offset reported on the QTP was not considered here because thermal offset measurements are not available for all sites, and the influence of the reversed thermal offset is expected to be minimal due to its small magnitude (the value was reported as -0.07 °C by Lin et al. (2015)). GPR data are from Cao et al. (2017b), and were measured in 2014 between late September and November using 100 and 200 MHz antennas. The GPR survey depth is from about 0.8 m to nearly 5 m depending on the active layer thickness. The data were carefully processed by removing opaque reflections, and evaluated using direct measurements. The ability of GPR data to detect permafrost relies on the strong dielectric contrast between liquid water and ice (Moorman et al., 2003). Consequently, it is more difficult to discern the presence of permafrost in areas with low soil moisture content because it weakens this contrast (Cao et al., 2017b). For this reason, the GPR data were only considered to indicate the presence of permafrost if an active layer thickness could be established.

In order to apply the permafrost presence or absence inventory more broadly, the degree of confidence in the data is estimated and provided in the inventory and in Table 1, although it is not used in this study. BH and SP provide direct evidence of permafrost presence or absence based on MGT and/or ground ice observations, and hence have high confidence (Cremonese et al., 2011). The data confidence derived from MAGST is classified based on temperature and the length of the observation period. The evaluated GPR survey result was considered to have medium confidence.

## 2.2 Topographical and climatological properties of the inventory sites

The slope and aspect for the inventory sites were derived from a DEM with 3 arc second spatial resolution, which is aggregated from the Global Digital Elevation Model version 2 (GDEM2) by averaging to avoid the noise in the original dataset (Cao et al., 2017a). The thermal state and spatial distribution of permafrost result from the long-term interaction of the climate and subsurface. Additionally, vegetation and snow cover play important roles in permafrost distribution by influencing the energy exchange between the atmosphere and the ground surface (Norman et al., 1995; Zhang, 2005). In this study, three climate variables were selected to test the representativeness of the inventory for permafrost map evaluation: mean annual air temperature (MAAT), mean annual snow cover days (MASCD), and the annual maximum normalized difference vegetation index ($NDVI_{max}$). The MAAT was obtained from Gruber (2012), it has a spatial resolution of 1 km and represents the reference period spanning 1961–1990. The MASCD, with a spatial resolution of about 500 m, was derived from a daily snow cover product developed by Wang et al. (2015) based on MODIS products (MOD10A1 and MYD10A1). To improve the comparison of MASCD, it was scaled to values between 0 and 1 by dividing the total days of a given year, and the mean MASCD during 2003–2010 was produced as a predictor. The annual maximum NDVI is from the MODIS/Terra 16-day Vegetation Index product (MOD13Q1, v006) which has a spatial resolution of 250 m. It was computed for each year between 2001–2017 to represent the approximate amount of vegetation, and then aggregated to a median value for the entire period to avoid sensitivity to extreme values. These climate variables were extracted for field site locations based on nearest-neighbor interpolation. The outline of the QTP is from Zhang et al. (2002), glacier outlines are from Liu et al. (2015) representing conditions in 2010, and lake data is provided by the Third Pole Environment Database.

## 2.3 Existing maps over the QTP

Table 2 gives a summary of the most widely used and recently developed permafrost maps over the QTP. In general, permafrost maps over the QTP could be classified as: (i) categorical, using categorical classification with different permafrost categories (e.g., continuous, discontinuous, sporadic, and island permafrost), or (ii) continuous, using a continuous probability or index with a range of [0.01–1] to represent the proportion of an area that is underlain by permafrost. The IPA map, which may be the most widely used categorical map, was compiled by assembling all readily available data on the characteristics and distribution of permafrost (Ran et al., 2012). The IPA map uses the "permafrost zone" to describe spatial patterns of permafrost, and the areas are divided into five categories based on the proportion of the ground underlain by permafrost: continuous (> 90%), discontinuous (50–90%), sporadic (10–50%), island (0–10%) and absent (0%). The most recent efforts were made by Zou et al. (2017) using the TTOP model (referenced as $QTP_{TTOP}$ map) forced by a calibrated (using station data) land surface temperature (or freezing and thawing indices) considering soil properties, and by Wu et al. (2018) based on the Noah land surface model (referenced as $QTP_{Noah}$ map) as well as gridded meteorological datasets, including surface air temperature, radiation, and precipitation. Although these two categorical maps are expected to be superior because they use the latest measurements and advanced methods, they were evaluated using limited and narrow distributed data (∼200 sites for the $QTP_{TTOP}$ map and 56 sites for the $QTP_{Noah}$ map). The $PZI_{global}$ map, which gives a continuous index value for permafrost distribution, is derived through a heuristic-empirical relationship with mean annual air temperature (MAAT) based on generalized linear models (Gruber, 2012). The model parameters are established largely based on the boundaries of continuous (PZI = 0.9 for MAAT = -8.0 °C) and island (PZI = 0.1 for MAAT = -1.5 °C) permafrost in the IPA map and do not use field observations. Gruber (2012) introduced two end-member cases for either cold (conservative or more permafrost) or warm (non-conservative or less permafrost) conditions, into the $PZI_{global}$ map to allow the propagation of uncertainty caused by input datasets and model suitability. The three cases or maps, referenced as $PZI_{norm}$, $PZI_{warm}$, and $PZI_{cold}$ maps, differ in the parameters used. Compared to the normal case, the cold and warm variants are derived by shifting PZI and MAAT at the respective limit by ± 5% and ± 0.5 °C, respectively. The $PZI_{global}$ map was partly evaluated for the QTP using rock glaciers, considered as indicators of permafrost conditions, based on remote sensing imagery (Schmid et al., 2015). However, rock glaciers, are absent in much of the QTP due to very low precipitation (Gruber et al., 2017).

## 2.4 Statistics and evaluation of permafrost distribution maps

In order to compare maps, it is important to understand the difference between extent of permafrost regions and permafrost area. Permafrost area refers to the quantified extent of area within a domain that is completely underlain by permafrost, whereas permafrost regions are categorical areas within a domain that are defined by the percent of land area underlain by permafrost. For example, extensive discontinuous permafrost is a region where, by definition, 50 to 90% of the land area is underlain by permafrost. In this discontinuous permafrost region of known area, the area actually underlain by permafrost is the permafrost area (Zhang et al., 2000).

To conduct the map evaluations against measurements with binary information (presence or absence), it was necessary to

develop classification aggregations for the existing maps. We argue that although the aggregation presented here simplifies the information available in these maps and may introduce uncertainty for further analyses, it is necessary in order to conduct inter-comparisons among them. For the IPA map, we consider the continuous and discontinuous permafrost zones to correspond to permafrost presence and the other zones (sporadic permafrost, island permafrost, and non-permafrost) to correspond to permafrost absence by using the proportion of ground underlain by permafrost of 50% as a threshold. This is consistent with the threshold of the PZI map described below. For the $QTP_{TTOP}$ and $QTP_{Noah}$ maps, the permafrost distribution was derived using simulated mean annual ground temperature (thermally defined). In these maps, areas are classified into three type: permafrost, seasonally frozen ground, and unfrozen ground. Here, we merge the areas of seasonally frozen ground and unfrozen ground to yield areas of permafrost absence. For the PZI maps, specified thresholds are required for both the extent of permafrost region and permafrost area. Following Gruber (2012), only the areas with PZI $\geq 0.01$ were selected for further analysis, permafrost regions were defined as where PZI $\geq 0.1$, and permafrost area was calculated as PZI multiplied pixel area. A value of 0.5 was used as the threshold of permafrost presence and absence (Boeckli et al., 2012; Azócar et al., 2017).

Maps were evaluated based on field evidence to produce accuracy measurements as follows (Wang et al., 2015) :

$$PCC_{PF} = \frac{PF_T}{PF_T + PF_F} \times 100\% \tag{1}$$

$$PCC_{NPF} = \frac{NPF_T}{NPF_T + NPF_F} \times 100\% \tag{2}$$

$$PCC_{tol} = \frac{PF_T + NPF_T}{PF_T + PF_F + NPF_T + NPF_F} \times 100\% \tag{3}$$

where $PF_T$ is the number of permafrost sites correctly classified as permafrost, while $PF_F$ is the number of permafrost sites incorrectly classified as non-permafrost. Similarly, $NPF_T$ is the number of permafrost-absent sites correctly classified as non-permafrost, and $NPF_F$ is the number of incorrectly classified non-permafrost sites. $PCC$ is the percentage of sites correctly classified, and the subscripts $PF$, $NPF$, and $tol$ indicate permafrost, non-permafrost, and total sites, respectively. To avoid the impact of unequal sample sizes in each of the two categories (presence and absence), the Cohen's kappa coefficient ($\kappa$), which measures inter-rater agreement for categorical items (Landis and Koch, 1977), was used for map evaluation:

$$\kappa = \frac{p_o - p_e}{1 - p_e} \tag{4}$$

where $p_e$ and $p_o$ are the probability of random agreement and disagreement, respectively, and can be calculated as

$$p_e = \frac{(PF_T + PF_F) \times (PF_T + NPF_F)}{(PF_T + PF_F + NPF_F + NPF_T)^2} \tag{5}$$

$$p_o = \frac{(NPF_F + NPF_T) \times (PF_F + NPF_T)}{(PF_T + PF_F + NPF_F + NPF_T)^2} \tag{6}$$

Cohen's kappa coefficient results are interpreted to mean excellent agreement for $\kappa \geqslant 0.8$, substantial agreement for $0.6 \leqslant \kappa < 0.8$, moderate agreement for $0.4 \leqslant \kappa < 0.6$, slight agreement for $0.2 \leqslant \kappa < 0.4$, and poor agreement for $\kappa < 0.2$.

# 3 Results and discussion

## 3.1 Evidence of permafrost presence or absence

There are a total of 1475 permafrost presence or absence sites contained in the inventory acquired from BH, SP, GST, and GPR methods (Figure 1). Among these, 1141 (77.4%) sites were measured by BH, 184 (12.5%) sites by GST, 144 (9.8%) sites by
GPR, and 6 (0.4%) sites by SP (Figure 1b). There are 1012 (68.6%) permafrost presence sites and 463 (31.4%) permafrost absence sites. The data cover a large area of the QTP (latitude: 27.73–38.96°N, longitude: 75.06–103.57°E) (Figure 1c) and a wide elevation range from about 1600 m to above 5200 m. However, the majority of sites (93.2%) are located between 3500 m and 5000 m. The inventory has an even distribution of aspects with 27.3% on the east slope, 27.9% on the south slope, 22.0% on the west slope, and 22.6% on the north slope. Most of the sites (96.1%) have slope angles less than 20° (Figure 1c).

Figures 1d, e, and f compare the distribution of three climate variables between the field sites and the entire QTP. The 1475 field sites have a narrower MAAT range (-10.5–15.7 °C with 25th percentile = -6.0 °C and 75th percentile = -3.8 °C) compared to the entire QTP which has a MAAT between -25.6 and 22.1 °C (25th percentile = -6.6 °C and 75th percentile = -0.41 °C), and only 1.5% sites located in the area with MAAT < -8 °C. However, the data (88.2%) were mostly found in the most sensitive MAAT range (from -8 to -2 °C) for permafrost presence or absence (Gruber, 2012; Cao et al., 2018). There is a slight bias in
the scaled MASCD coverage. Few measurements (7.5%) were located in areas of high scaled MASCD (> 0.20) due to the associated harsh climate and inconvenient access. The $NDVI_{max}$ at field evidence sites have a wide coverage for the QTP with the range of 0.05–0.88. The higher mean $NDVI_{max}$ for field sites (0.44 at the sample sites and 0.37 for the QTP) is due to the fact that measurements were normally collected in flat areas with relatively dense vegetation cover. These results suggest that the evaluation presented in this study are representative of most of the QTP but may have more uncertainty in steep and
regularly snow-covered regions.

## 3.2 Evaluation and comparison of existing maps

The new inventory was used to evaluate existing permafrost maps derived with different methods (Table 2). In general, these permafrost maps showed different performances, including slight agreement for the IPA map, fair agreement for the $PZI_{warm}$ map, moderate agreement for the $QTP_{Noah}$, $PZI_{norm}$, $PZI_{cold}$, and $QTP_{TTOP}$ maps, with a wide spread of $\kappa$ from 0.21 to 0.58. The
high $PCC_{PF}$ together with low $PCC_{NPF}$ for the $QTP_{Noah}$, $PZI_{cold}$, and $QTP_{TTOP}$ maps indicate permafrost is overestimated by them, while the IPA, $PZI_{warm}$, and $PZI_{norm}$ maps underestimated the permafrost over the QTP. Despite the small permafrost area bias for the $QTP_{TTOP}$ and $QTP_{Noah}$ maps caused by different QTP boundaries, lake, and glacier datasets used, the range of estimated permafrost region (1.42–1.84 $\times$ $10^6$ km$^2$, or 30% difference) and area (0.76–1.25 $\times$ $10^6$ km$^2$, or 64.4% difference) are extremely large (Figure 2).

Among the categorical maps, the $QTP_{TTOP}$ map achieved the best performance for permafrost distribution over the QTP with the highest $\kappa$ (0.58, moderate agreement) and $PCC_{tol}$ (82.8%), however, caution should be taken when interpolating the map. The $QTP_{TTOP}$ map was derived based on MODIS land surface temperature with temporal coverage of 2003–2012 (Zou et al., 2017). Though the MODIS land surface temperature time-series gaps caused mainly by clouds were filled using the

Harmonic Analysis Time-Series (HANTS) algorithm (Prince et al., 1998), the surface conditions, especially vegetation and snow cover, were ignored. In this case, land surface temperature is underestimated in high or dense vegetation areas because it comes from the top of the vegetation canopy, and is overestimated in snow-covered areas where the cooling effects of snow are not considered. As a consequence, permafrost is likely overestimated in areas of high or dense vegetation and underestimated in regularly snow-covered areas. While the QTP$_{\text{Noah}}$ map performed slightly better (2.5 % higher) for permafrost area than the QTP$_{\text{TTOP}}$ map, it suffer from considerable underestimation of non-permafrost area (12.7% lower for $PCC_{NPF}$). Although the QTP$_{\text{Noah}}$ map was derived using a coupled land surface model (Noah), the poorer performance, especially for non-permafrost area ($PCC_{\text{NPF}} = 49.5\%$), is likely caused by the coarse-scale forcing dataset ($0.1°$ resolution or $\sim10$ km) and by the uncertainty in the soil texture dataset (Chen et al., 2011; Yang et al., 2010). It is not surprising that the IPA map has slight agreement ($\kappa = 0.21$) because fewer observations were compiled and the methods used were more suitable for high latitudes (Ran et al., 2012).

For the PZI map, the PZI$_{\text{norm}}$ and PZI$_{\text{cold}}$ maps were found to be in moderate agreement ($\kappa = 0.56$ for the PZI$_{\text{norm}}$ map and 0.55 for the PZI$_{\text{cold}}$ map) with in-situ measurements, and performed slightly worse than the QTP$_{\text{TTOP}}$ map. The poor performance of the PZI$_{\text{warm}}$ map and underestimation of the PZI$_{\text{norm}}$ map indicated that permafrost over the QTP is more prevalent than most of the other regions even though the climate conditions, especially the MAAT, are similar. This is likely because of the high soil thermal conductivity due to coarse soil and the cooling effects of minimal snow (Zhang, 2005). Large differences of permafrost region ($0.42 \times 10^6$ km$^2$, or 25% of the normal case) and area ($0.49 \times 10^6$ km$^2$, or 49% of the normal case) were found for the three cases of the PZI$_{\text{global}}$ map, though the upper and lower bounds only changed about 5% for the PZI and $\pm$ 0.5 °C for the MAAT. The MAAT used in the PZI$_{\text{global}}$ map was statistically downscaled from reanalysis based on the lapse rate derived from NCEP upper-air (pressure level) temperatures. The land surface influences on surface air temperature, such as cold air pooling, were ignored (Cao et al., 2017a). This is important as winter inversions are excepted to be common due to the prevalent mountains over the QTP. In other words, permafrost may be underestimated in valleys due to the overestimated MAAT.

Spatially, the non-permafrost areas of the southeastern QTP are well represented in all maps, while misclassification is prevalent in areas near the boundary between permafrost and non-permafrost and in spatially highly variable landscapes such as the sources of the Yellow River (Figure 2). This is because the permafrost spatial patterns in these areas are not only controlled by medium- to large-scale climate conditions (e.g., MAAT), which are described by the models used, but also strongly influenced by various local factors such as peat layers, thermokarst, soil moisture, and hydrological processes. The IPA and PZI$_{\text{warm}}$ maps showed a fit that is good only in some areas (e.g., relatively colder areas for the IPA map and southeastern for the PZI$_{\text{warm}}$ map) based on the in-situ measurements, and may not represent the permafrost distribution patterns well for the other areas beyond the measurements.

## 4 Conclusions

We compiled an inventory of evidence for permafrost presence or absence using 1475 field sites obtained based on diverse methods over the QTP. With a wide coverage of topography (e.g., elevation and slope aspect) and climate conditions (e.g.,

surface air temperature and snow cover), the inventory gives a representative baseline for site-specific permafrost occurrence.

The existing permafrost maps over the QTP were evaluated and inter-compared using the inventory of ground-based evidence, and they showed a wide range of performance with the $\kappa$ from 0.21 to 0.58 and overall classification accuracy of about 55–83%. The misclassification is prevalent in areas near the boundary between permafrost and non-permafrost and in spatially highly variable landscapes. This highlights the need for improved mapping methods that consider more controlling factors at both medium-large and local scales. The QTP$_{TTOP}$ map is recommended for representing permafrost distribution over the QTP based on our evaluation. Additionally, the PZI$_{norm}$ and PZI$_{cold}$ maps are similar to one another and are valuable alternatives for describing a permafrost zonation index over the QTP. The inadequate sampling in steep and regularly snow-covered areas is expected to result in higher uncertainty for map evaluation and requires further investigation using systematic samples.

*Data availability.* Inventory of permafrost presence or absence is partly available as supplement, the other evidence sites not listed are available from the authors upon request.

*Author contributions.* BC carried out this study by organizing the inventory of permafrost presence or absence evidence, analyzing data, performing the simulations and by structuring as well as writing the paper. TZ guided the research. QW, YS, LZ, and DZ contributed to organize the permafrost presence or absence dataset.

*Competing interests.* The authors declare that no competing interests are present.

*Acknowledgements.* The authors would like to thank the Editor Peter Morse, two anonymous reviewers, Stephan Gruber, and Kang Wang for their constructive suggestions. We thank Nicholas Brown for improving the writing of earlier manuscript. We thank Zhuotong Nan and Xiaobo Wu for providing the QTP$_{Noah}$ map. This study was supported by the Strategic Priority Research Program of Chinese Academy of Sciences (XDA20100103, XDA20100313), the National Natural Science Foundation of China (41871050, 41801028), partly by the Fundamental Research Funds for the Central Universities (lzujbky_2016_281, 862863). We thank CMA (http://cdc.cma.gov.cn/) for providing the surface air and ground surface temperatures, the GDEM2 dataset is downloaded from United States Geological Survey (http://gdex.cr.usgs.gov/gdex/), the NDVI datasets are derived and processed in the Google Earth Engine, glacier inventory is provided by the Environmental and Ecological Science Data Center for West China (http://westdc.westgis.ac.cn/), and the lake inventory is from the Third Pole Environment Database (http://www.tpedatabase.cn).

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

**Table 1.** Classification algorithm of in-situ permafrost presence or absence evidence from various methods

| Method | Indicator | Survey depth | Permafrost | Confidence degree |
|--------|-----------|--------------|------------|-------------------|
| BH | MGT $\leqslant$ 0 °C | meters to about 20 m | presence | high |
| SP | ground ice presence | about 1.0–2.5 m | presence | high |
| GST | MAGST $\leqslant$ -2 °C & observations $\geqslant$ 3 | 0.05 or 0.1 m | presence | medium |
| | MAGST $\leqslant$ -2 °C & observations $<$ 3 | | presence | low |
| | MAGST $>$ -2 °C & $MAGST + TO_{\max} \leqslant$ 0 °C | | presence | low |
| | MAGST $<$ 0 °C & $MAGST + TO_{\max} >$ 0 °C | | ambiguous | – |
| | MAGST $>$ 0 °C | | absence | medium |
| GPR | active layer thickness could be estimated | about 0.80–5.0 m | presence | medium |

BH = borehole temperature, SP = soil pit, GST = ground surface temperature, GPR = ground-penetrating radar, MGT = mean ground temperature, and MAGST = mean annual ground surface temperature. $TO_{\max}$, the maximum thermal offset under natural conditions reported for the QTP, is 0.79 °C. Ambiguous means the data is not sufficient to determine permafrost conditions and is not included in the inventory.

**Table 2.** Summary and evaluation of existing permafrost maps over the Qinghai-Tibet Plateau

| Name | IPA | QTP$_{TTOP}$ | QTP$_{Noah}$ | PZI$_{norm}$ | PZI$_{warm}$ | PZI$_{cold}$ |
|---|---|---|---|---|---|---|
| Year | 1997 | 2017 | 2018 | 2012 | 2012 | 2012 |
| Method | – | semi-physical model | physical model | heuristic GLM | heuristic GLM | heuristic GLM |
| Classification criteria | categorical | categorical | categorical | continuous | continuous | continuous |
| Scale | 1:10,000,000 | ∼1 km | 0.1° (∼10 km) | ∼1 km | ∼1 km | ∼1 km |
| $PCC_{PF}$ [%] | 46.6 | 93.9 | 96.4 | 76.6 | 35.3 | 94.3 |
| $PCC_{NPF}$ [%] | 79.8 | 58.6 | 45.9 | 82.6 | 98.5 | 54.0 |
| $PCC_{tol}$ [%] | 57.0 | 82.8 | 80.7 | 78.5 | 55.1 | 81.7 |
| $\kappa$ | 0.21 | 0.58 | 0.52 | 0.56 | 0.36 | 0.55 |
| PF region [$10^6$ km$^2$] | 1.63 | – | – | 1.68 | 1.42 | 1.84 |
| PF area [$10^6$ km$^2$] | – | $1.06 \pm 0.09$ | 1.13 | 1.00 | 0.76 | 1.25 |
| Reference | Brown (1997) | Zou et al. (2017) | Wu et al. (2018) | Gruber (2012) | Gruber (2012) | Gruber (2012) |

Evaluations are conducted using 1475 in-situ measurements of permafrost presence or absence. GLM = generalized linear model, PF = permafrost. Norm (normal), warm, and cold means different cases and assumptions of parameters for permafrost distribution simulations in the PZI$_{global}$ map, details are from Table 1 of Gruber (2012). The continuous classification criteria means the permafrost spatial patterns is compiled or present as continuous value with a range of [0.01–1], e.g., permafrost zonation index in the PZI maps.

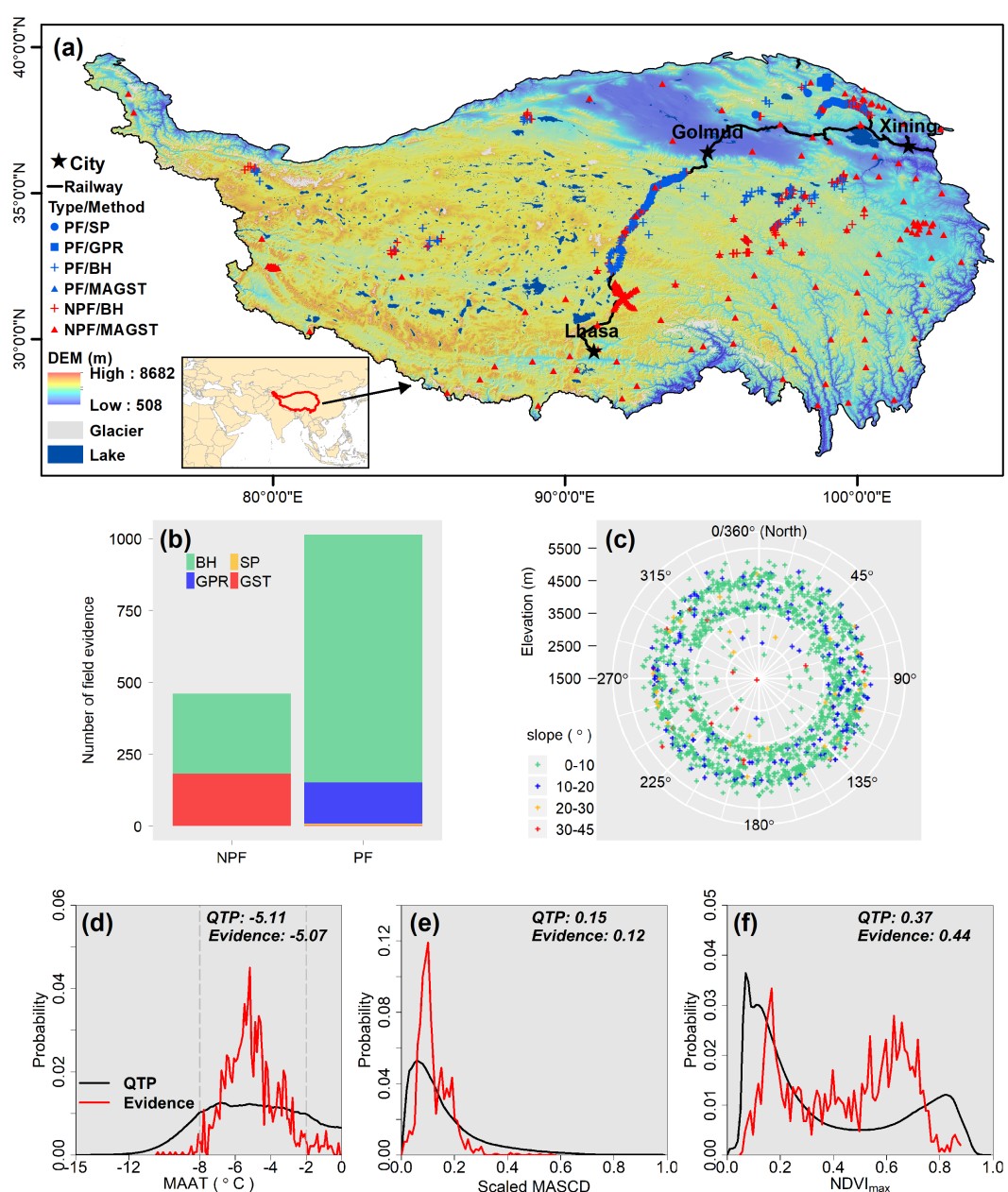

**Figure 1.** (a) The location of the QTP, and in-situ permafrost presence (PF) or absence (NPF) evidence distribution over the QTP, superimposed on the background of digital elevation model (DEM) with a spatial resolution of 30 arc second. (b) Number of field evidence located in NPF and PF regions. SP means soil pit, GPR refers ground-penetrating radar, BH stands field evidence measured by borehole drilling, and MAGST is mean annual ground surface temperature. (c) Distribution of field evidence in terms of elevation (radius), slope (colored), and aspect (0/360° represents North). Distributions of (d) mean annual air temperature (MAAT), (e) scaled mean annual snow cover days (MASCD), and (f) annual maximum NDVI (NDVI$_{max}$) for field evidence (red line) comparing to the entire QTP (black line). Numbers in (d), (e), and (f) are mean values. Only the sites with MAAT < 0 °C, which is the precondition for permafrost presence, were present in (d).

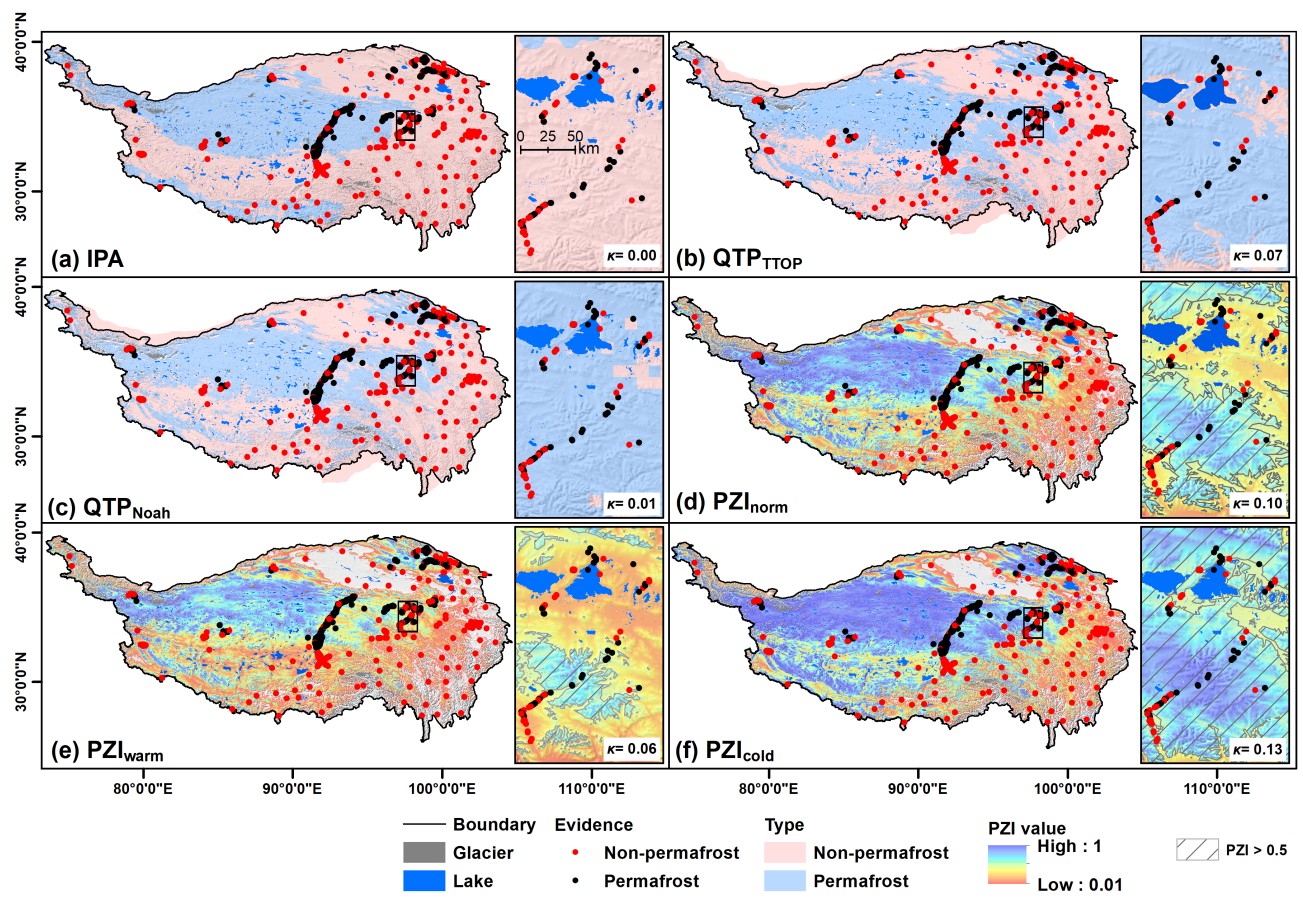

**Figure 2.** The permafrost classification results at in-situ evidence sites shown on the (a) IPA, (b) QTP$_{TTOP}$, (c) QTP$_{Noah}$, (d) PZI$_{norm}$, (e) PZI$_{warm}$, and (f) PZI$_{cold}$ maps. The Cohen's kappa coefficient ($\kappa$), was derived from the selected spatially highly variable landscapes (marked by black box) with 106 evidence sites. All the maps are re-sampled to the unprojected grid of SRTM30 DEM with a spatial resolution of 30 arc second (∼1 km) to avoid maps bias caused by different resolutions, geographic projection, and format. The boundary of QTP used in this study is marked by black line. Categorical classification is used for the QTP$_{TTOP}$, QTP$_{Noah}$, and IPA maps, while continuous PZI was present for the PZI$_{norm}$, PZI$_{warm}$, PZI$_{cold}$ maps. The blank parts in the PZI maps are areas with PZI < 0.01.