# Peer review of "Brief Communication: Evaluation and inter-comparisons of permafrost map over the Qinghai-Tibet Plateau based on inventory of in-situ evidence"

_The Cryosphere, 2018_

## Referee Comment (RC1) · Anonymous Referee #1 · 17 Oct 2018

Permafrost maps were released by various institutes or research teams during the past several decades. They used modeling, statistical, and other mapping techs. Basically, the maps were evaluated during processing. However, the inter-comparison, what this study was done, is required for better understanding. This study collected more than a thousand samples over the QTP. The results of this study would be useful for future permafrost studies on the QTP and broad interest to the permafrost communities.

The manuscript, however, requires a bit more work before it is acceptable for publication. For the most part, the manuscript is well written but some editing is required to improve language and increase clarity. There are a few places in the manuscript where

more explanation would be helpful.

Although I have made a few comments here that I hope the authors will find useful, dealing with them may not take too much time. The authors should thoroughly proofread the revised manuscript before submission or invite a native speaker in permafrost communities to improve the language. I am willing to review the revised paper.

**Major:**

- **Unclear description and logic (to the following results) in the Data and Methods section.**

  The authors used four methods to classify permafrost or not. However, it's not enough for understanding, although this paper is a short communication.

  – How deep are generally for boreholes and soil pits? 1 m, 5 m?

  – It looks like this study used only $MAGST + TO_{Max} \leq 0$ as the standard. In your results, you only talked about the sites considered as permafrost. I am not sure whether these classifications (P2, L25-29) are necessary.

  – What is kind of antennas generally used in GPR survey? Also, how deep is accessed?

  – In section 2.3, you used DEM (3 arc second), MAAT (1 km), MASCD (∼500 m), and NDVI (∼250 m). I guess you extract those variables for each site in your inventory. Is it? You also said (P5, L2-4) "Where original field evidence of permafrost presence/absence is located within the same grid cell (30 arc-sec, 1 km), they were aggregated based on their major value. For a grid with one permafrost site and one non-permafrost site, the nearer site from the grid center was used to represent the grid." (actually, these sentences

should be moved to section 2.3). Why did you have to aggregate these in-situ data to 1 km? How did you deal with DEM, MASCD, and NDVI? Did you upscale DEM, MASCD, and NDVI to 1 km? I guess you were going to avoid conflict sites (permafrost and non-permafrost) in the same pixel. Is it? When you extracted values from different spatial resolution datasets, even if there are probably few sites in the same pixel at 1 km resolution, however, there still are three spatial datasets with higher resolution, which might bring different snow, topography, and vegetation condition to your sites. In fact, there might be different ground thermal states under the same climate and vegetation condition because of different soil wetness, soil properties, and so on. Overall, I don't think the aggregation is necessary. Furthermore, how did you compare with the maps with different spatial scale, e.g., $QTP_{Noah}$ map is 10 km. Those issues were confusing and should be clarified.

- **Misleading indicators.**

$PCC_{PF}$, $PCC_{NPF}$, and $PCC_{tol}$ were used to quantify the classification accuracies of permafrost maps. To my sense, $PCC_{PF}$ and $PCC_{NPF}$ are not useful and may be misleading. When the map over-presents permafrost (i.e., much colder), $PCC_{PF}$ would be extremely close to 100%. Can we say this is much better? Vice versa. Thus, the description in Section 3.2 could be misleading, at least to me, and should be more cautious. I suggest removing those parts. Meanwhile, do you consider the effect of the different sample volume? Because in your in-situ sites pool, number of sites with permafrost is twice as large as the sites without permafrost.

- **More discussion?**

This study found different performance in permafrost maps. It's better to discuss a little bit more about the sources of bias, such as different MAAT products. More discussion on the possible sources of the revealed differences would enhance

the scientific significance. Meanwhile, it also is useful for the future permafrost map updating.

**Specific:**

- P1, title: the title could be "Ground-based evaluation and inter-comparisons of permafrost maps over the Qinghai-Tibet Plateau"?

- P1, L3: the number, 1475, might be misleading although you collected. Because you aggregated to 1040, which excluded about 400 sites. Add a comma to 1040/1475 for consistency.

- P2, L1: "hemisphere" $->$ hemispheric ?

- P2, L10: "2000" $->$ "the 2000s"?

- P2, L16: insert "survey" after GPR.

- P2, L25-29: Where is so-call "high certainty" for permafrost classification? Meanwhile, it looks like this study used only $MAGST + TO_{Max} \leq 0$ as the standard. I am not sure whether these classifications are necessary. If necessary, the authors should clarify.

- P3, L1: The authors should briefly clarify what kind of antennas were used and how deep is accessible.

- P3, Section 2.2: It's worth to note what climate data were used in $QTP_{TTOP}$ and $QTP_{Noah}$ maps. Both used the data merged MODIS temperature products and station data?

- P3, L6: "(1)' $->$ "(i)"

- P3, L7: "(2)' − > "(ii)"

- P3, L11: "... the temperature at ..." − > "...the mean annual temperature at..."

- P4, L5: " ... outline of QTP ..." − > "...outline of the QTP..."

- P4, L24: Is the calculation of "Cohen's kappa coefficient" too complicated? If not, please put equation(s) here and indicate what a high k means. Is there some threshold to roughly classify good, fair, or others?

- P5, L12-13: What's Qxx?

- P5, L15: Cao et al. (?), missing year.

- P6, L14: why is "-3 to -4 C"? Generally, -4 to -3 C?

---

## Referee Comment (RC2) · Anonymous Referee #2 · 18 Oct 2018

The manuscript presents a useful contribution for understanding performance of different permafrost maps at QTP. The aim of the study, methods and presented results are relatively clear, however, several parts of the text need to be clarified and part of the methods needs to be slightly extended. The manuscript has to be proofread for language and use of several terms in the manuscript can be improved. I have listed a number of specific comments below, which should improve the clarity of the text. Authors should find the comments straightforward to implement.

Specific comments:

Page 1, line 4: change "overall accuracy of about" to "overall accuracy between"

[Figure]

Page 1, line 5: omit "extremely large". The areas are matter of scale and don't need to be evaluated in this case. It is also not clear how this part of the sentence relates to the beginning where comparison to in-situ measurements is discussed.

Page 1, line 6: How do you define "fragile landscapes"?

Page 2, lines 4-5: What is a large enough dataset? I assume that the evaluation datasets were large enough for the publications to be published. In the next sentence, "This would weaken their applications" sounds as the datasets were inappropriate. I would change the formulations of the both sentence to more positive. For instance: "The new larger dataset can be used to improve evaluations of the existing datasets, which would further improve their applications . . ."

Page 2, line 16: The word evidence is used at many places in the manuscript. I'm not sure that its use is correct. It could be replaced by "information" in this case and maybe just a "validation site" elsewhere in the manuscript.

Page 2, line 18: The use of word "confidence" shall be used instead of "certainty" also further in the manuscript.

Page2, line 25: What are your criteria to define confidence (certainty) classes medium and low? How are these classes used further in the manuscript?

Page 3, lines 4-5: How do you define a clear permafrost reflection? The exact criteria for selection of GPR sites should be presented.

Page 3, line 9: The IPA map shows extent of four permafrost zones and is therefore not a binary map. Present here how did you convert it in to binary map showing permafrost presence and absence.

Page 3, line 16: Please explain here how PZIcold, PZIwarm and PZInorm were derived by Gruber (2012) and what is difference between them.

Page 3, consider moving 2.3 section before 2.2 because it is in my opinion logical

continuation of the inventory of permafrost validation sites. Also consider changing the section title to "Topographical and climatological properties of the inventory (or permafrost validation) sites"

Page 3, line 32: What are you referring to with "(about 500m)"?

Page 4, line 9: Please consider extending the explanation about the difference between permafrost area and permafrost region. This concept is difficult to understand by broader permafrost community. Maybe introduce the concept of scale and ground coverage by permafrost.

Page 4, lines 26-27: Restructure the sentences. It sounds as because of your permafrost absence/absence classification, you have 1475 sites. I assume that this is because of your site selection criteria.

Page 5, line 3: "were aggregated based on their major value". Maybe replace with "the majority value was assigned to aggregated sites.

Page 5, line 15: More appropriate term for "band" would be "range". What exactly does the word "sensitive" refer to?

Page 5, line 31: Did you mean QTPTTOP instead of PZITTOP?

Page6, line 10: Again, how exactly is fragile landscape defined?

---

## Author Comment (AC1) · 31 Oct 2018

**Response to Anonymous Referee #1**

**Anonymous Referee #1**

The authors would like to thank the reviewer for the constructive feedback, and the thorough assessment of the manuscript. Below we provide a point-to-point response to each comment, reviewer comments are given in black, responses are given in blue. Additionally, we have included details of how we intend to address these changes in a revised submission.

Permafrost maps were released by various institutes or research teams during the past several decades. They used modeling, statistical, and other mapping techs. Basically, the maps were evaluated during processing. However, the inter-comparison, what this study was done, is required for better understanding. This study collected more than a thousand samples over the QTP. The results of this study would be useful for future permafrost studies on the QTP and broad interest to the permafrost communities.

The manuscript, however, requires a bit more work before it is acceptable for publication. For the most part, the manuscript is well written but some editing is required to improve language and increase clarity. There are a few places in the manuscript

where more explanation would be helpful.
Response: The language of revised manuscript will be carefully checked.

Although I have made a few comments here that I hope the authors will find useful, dealing with them may not take too much time. The authors should thoroughly proofread the revised manuscript before submission or invite a native speaker in permafrost communities to improve the language. I am willing to review the revised paper.

**Major:**

- **Unclear description and logic (to the following results) in the Data and Methods section.**
  The authors used four methods to classify permafrost or not. However, it's not enough for understanding, although this paper is a short communication.

  – How deep are generally for boreholes and soil pits? 1 m, 5 m?
  Response: In general, the borehole depths vary from meters to hundred meters. In this study, we used the mean annual ground temperature from boreholes, which also varies from several meters to about 20 m, to identify permafrost presence. Number of samples measured from soil pits was small (6 samples) due to the prevalent coarse soil, and their depths are between less than 1 m to about 2.5 m. In the revised manuscript, we will add

  *"In this study, we used the mean annual ground temperature (MAGT) measured from boreholes, which varies from meters to about 20 m to identify permafrost presence or absence. Due to the prevalent coarse soil, SP was only applied in*

*areas possible, and the depth is from less than 1 meter to about 2.5 m."*

to clarify.

Additionally, the survey depth of all the methods will be summarized in Table A1 (see below)

**Table *A1.*** *Classification algorithm of in-situ permafrost presence or absence evidence from various methods*

| Method | Indicator | Survey depth | Permafrost | Confidence degree |
|---|---|---|---|---|
| BH | $MAGT \leqslant 0$ | meters to about 20 m | presence | high |
| SP | ground ice presence | about 1.0–2.5 m | presence | high |
| GST | $MAGST \leqslant \text{-}2\,°C \;\&\; observations \geqslant 3$ | 0.05 or 0.1 m | presence | medium |
| | $MAGST \leqslant \text{-}2\,°C \;\&\; observations < 3$ | | presence | low |
| | $MAGST > \text{-}2\,°C \;\&\; MAGST + TO_{max} \leqslant 0\,°C$ | | presence | low |
| | $MAGST < 0\,°C \;\&\; MAGST + TO_{max} > 0\,°C$ | | ambiguous | – |
| | $MAGST > 0\,°C$ | | absence | medium |
| GPR | clear permafrost reflection | about 0.80–5.0 m | presence | medium |

*BH = borehole temperature, SP = soil pit, GST = ground surface temperature, and GPR = ground-penetrating radar. $TO_{max}$, the maximum thermal offset under natural conditions reported for the QTP, is 0.79 °C.*

– It looks like this study used only $MAGST + TO_{max} \leqslant 0$ as the standard. In your results, you only talked about the sites considered as permafrost. I am not sure whether these classifications (P2, L25-29) are necessary.

Response: As mentioned by Referee #2 (specific comment Page 2, line 18), the word "certainty" is changed to "confidence". Yes, the confidence classifications were not further used in this manuscript, but only present in the inventory as supplement. Since the inventory may be useful for other researches, we would keep the classification in the inventory and move the classification description into the Appendix A (see below). We hope you agree.

The classification algorithm of confidence degree largely follows Cremonese et al. (2011) and could be summarized as

*"Appendix A: Classification algorithm of in-situ permafrost presence or absence evidence*
*For board use of the permafrost presence or absence inventory, the data confidence degree was provided (Table A1). BH and SP provide direct evidence of permafrost presence or absence based on MAGT and/or ground ice observations, and hence have high confidence (Cremonese et al., 2011). The data confidence derived from MAGST is classified based on temperature and the length of the observation period. The evaluated GPR survey result was considered as medium confidence."*

– What is kind of antennas generally used in GPR survey? Also, how deep is accessed?

Response: The GPR survey was conducted using 100 and 200 MHz antennas and evaluated using direct measurements (e.g., mechanical probing, soil temperature, and soil pits) (Cao et al. , 2017). The survey depth was from about 0.8

to near 5 m depending on the active layer thickness. In the revised manuscript, authors will add

*"Here, GPR data from (Cao et al. , 2017) are measured using 100 and 200 MHz antennas depending on the active layer thickness. The GPR survey depth is from about 0.8 to near 5 m, and the data are considered as indicating the presence of permafrost only if an active layer thickness (or a clear permafrost reflection) could be established."*

to clarify.

– In section 2.3, you used DEM (3 arc second), MAAT (1 km), MASCD (∼500 m), and NDVI (∼250 m). I guess you extract those variables for each site in your inventory. Is it?
Response: First of all, it is moved to Section 2.2 as Referee #2 suggested. Yes. We extract these variables to sample sites using nearest interpolation. We will add

*"These climate variables were extracted to in-situ sites/plots based on nearest interpolation."*

to clarify.

You also said (P5, L2-4) "Where original field evidence of permafrost presence/absence is located within the same grid cell (30 arcsec, 1 km), they were aggregated based on their major value. For a grid with one permafrost site and one non-permafrost site, the nearer site from the grid center was used to

represent the grid." (actually, these sentences should be moved to section 2.3). Why did you have to aggregate these in-situ data to 1 km?

Response: Yes, they are located within the same grid cell of unprojected SRTM30 with a spatial resolution of 30 arcsec. The aggregation will be deleted in the revised manuscript.

How did you deal with DEM, MASCD, and NDVI? Did you upscale DEM, MASCD, and NDVI to 1 km? I guess you were going to avoid conflict sites (permafrost and non-permafrost) in the same pixel. Is it?

Response: No. The 3 arcsec DEM was used to simulate the slope and aspect for the in-situ sites. The MASCD, NDVI, and MAAT are used here to explore the representative of the inventory, and they are extracted to the sites based on nearest interpolation.

When you extracted values from different spatial resolution datasets, even if there are probably few sites in the same pixel at 1 km resolution, however, there still are three spatial datasets with higher resolution, which might bring different snow, topography, and vegetation condition to your sites. In fact, there might be different ground thermal states under the same climate and vegetation condition because of different soil wetness, soil properties, and so on. Overall, I don't think the aggregation is necessary.

Response: Yes, we agree. In the revised manuscript, we will omit the evidence aggregation and conduct the evaluation with all the 1475 sites. Please note that, some statistics may be slightly different by using the original 1475 evaluation sites/plots.

Furthermore, how did you compare with the maps with different spatial scale, e.g., $QTP_{Noah}$ map is 10 km. Those issues were confusing and should be

clarified.

Response: The evaluation was conducted at the sites we collected. Permafrost presence and absence information at evaluation sites was extracted to the evidence based on nearest from different maps. In Section 2.4 Statistics and evaluation of permafrost distribution maps, we will add

*"The permafrost and absence information was extracted to in-situ sites, and..."*

to clarify.

• Misleading indicators.

$PCC_{PF}$, $PCC_{NPF}$, and $PCC_{tol}$ were used to quantify the classification accuracies of permafrost maps. To my sense, $PCC_{PF}$ and $PCC_{NPF}$ are not useful and may be misleading. When the map over-presents permafrost (i.e., much colder), $PCC_{PF}$ would be extremely close to 100%. Can we say this is much better? Vice versa. Thus, the description in Section 3.2 could be misleading, at least to me, and should be more cautious. I suggest removing those parts.

Response: Yes, we agree that $PCC_{PF}$ and $PCC_{NPF}$ are somehow misleading when we look at them separately without due care. On the other hand, these two indictors would be useful if they are jointly interpolated. As you mentioned, the high $PCC_{PF}$ together with low $PCC_{NPF}$ indicate the map over-presents permafrost. This information could not be indicated by either kappa coefficient nor $PCC_{tol}$. For this reason, we would keep these three indicators. To avoid the misunderstanding, the $PCC_{PF}$ and $PCC_{NPF}$ are interpolated together throughout the manuscript, and the over- or less-presents permafrost was also present. Additionally, the $PCC_{tol}$ in Figure 2 will be deleted to avoid confusion. In

the revised manuscript, we will add

*"The high PCC$_{PF}$ together with PCC$_{NPF}$ for the IPA, QTP$_{Noah}$, PZI$_{cold}$, and QTP$_{TTOP}$ maps indicate permafrost is over-presented by them, while the PZI$_{warm}$ and PZI$_{norm}$ showed underestimated the permafrost over the QTP."*

Meanwhile, do you consider the effect of the different sample volume? Because in your in-situ sites pool, number of sites with permafrost is twice as large as the sites without permafrost.
Response: Yes, the kappa coefficient, *"which measures inter-rater agreement for categorical items"*, was introduced here as the major indicator for map evaluation as it could largely *"avoid the impact of uneven distribution of sample numbers for permafrost presence and absence"*.

- More discussion?
This study found different performance in permafrost maps. It's better to discuss a little bit more about the sources of bias, such as different MAAT products. More discussion on the possible sources of the revealed differences would enhance the scientific significance. Meanwhile, it also is useful for the future permafrost map updating.
Response: Yes, we agree. Our previous manuscript had partly discussed the bias from inputs for the QTP$_{Noah}$ and IPA maps. In the revised manuscript, we will enhance this part, and the inputs bias will be discussed for each map as below:

QTP$_{Noah}$ map: *"Though the QTP$_{Noah}$ map was derived using coupled land surface model (Noah), the relatively worse performance, especially for non-permafrost area (PCC$_{NPF}$ = 45.9%), is likely caused by inputting coarse-scale forcing dataset (0.1° resolution or ~10 km) (Chen et al., 2011) and by the*

*uncertainty of soil texture dataset (Yang et al., 2010)."*

IPA map: *"It is not surprising that the IPA map has fair agreement (k = 0.32) as less observations were compiled and the method used are more suitable for high latitudes (Ran et al., 2012)."*

$QTP_{TTOP}$ map: *"The $QTP_{TTOP}$ map was derived based on MODIS land surface temperature with different temporal coverage of 2003–2012 (Zou et al., 2017). Though the MODIS land surface temperature time-series gaps caused mainly by cloud were filled using the Harmonic Analysis Time-Series (HANTS) algorithm (Prince et al., 1998), the surface conditions, especially vegetation and snow cover, were ignored. In this case, land surface temperature is underestimated in high and/or dense vegetation area as it comes from the top of vegetation canopy, and is overestimated in snow covered area due to the cooling effects of snow is not considered. As a consequence, permafrost is likely overestimated in high and/or dense vegetation area and underestimated in regular snow-covered area."*

$PZI_{global}$ map: *"The MAAT used in the $PZI_{global}$ map was statistical downscaled based on the lapse rate from the upper-air (or pressure level) temperature of NCEP, but the influences of land surface on surface air temperature, such as cold air pooling, was ignored (Cao et al., 2017). This is important as winter inversion is excepted to be common due to the prevalent mountains over the QTP. In other words, permafrost may be underestimated in valleys due to the overestimated MAAT."*

**Specific:**

- P1, title: the title could be "Ground-based evaluation and inter-comparisons of permafrost maps over the Qinghai-Tibet Plateau"?
  Response: We will change the title to

  *"Evaluation and inter-comparisons of permafrost map over the Qinghai-Tibet Plateau based on inventory of in-situ evidence".*

  As the study also provided the first inventory of permafrost presence or absence over the Qinghai-Tibet Plateau based on in-situ evidence, authors would like to reflect the inventory in the title. I hope you agree.

- P1, L3: the number, 1475, might be misleading although you collected. Because you aggregated to 1040, which excluded about 400 sites. Add a comma to 1040/1475 for consistency.
  Response: The aggregation part is removed, and evaluation was conducted using all the data.

- P2, L1: "hemisphere" –> hemispheric ?
  Response: Corrected.

- P2, L10: "2000" –> "the 2000s"?
  Response: Corrected.

- P2, L16: insert "survey" after GPR.
  Response: Done.

- P2, L25-29: Where is so-call "high certainty" for permafrost classification? Meanwhile, it looks like this study used only $MAGST + TO_{max} \leqslant 0$ as the standard. I am not sure whether these classifications are necessary. If necessary, the authors should clarify.
Response: As the Referee #2 mentioned, the word "certainty" is changed to "confidence". The evidence derived from BH and SP is considered as high confidence as they provide direct information, such as mean annual ground temperature or ground ice presence. Yes. To determine permafrost presence or absence, only the function of $MAGST + TO_{max} \leqslant 0$ was used. The confidence classifications were not used in this manuscript, but only present in the inventory as supplement. Since the inventory may be used for other related studies (e.g., permafrost simulation evaluation), and the confidence information would be useful for further selecting the data, we would keep the classification in the inventory and move the classification description into the Appendix A. Please also see our response to the major comments of "Unclear description and logic".

- P3, L1: The authors should briefly clarify what kind of antennas were used and how deep is accessible.
  Response: In the revised manuscript, the author will add

  *"Here, GPR data from Cao et al., (2017) are measured using 100 and 200 MHz antennas depending on the active layer thickness. The GPR survey depth is from about 0.8 to near 5 m, and the data are considered as indicating the presence of permafrost only if only an active layer thickness (or a clear permafrost reflection) could be established."*

  to clarify.

- P3, Section 2.2: It's worth to note what climate data were used in QTP$_{TTOP}$ and QTP$_{Noah}$ maps. Both used the data merged MODIS temperature products and station data?
  Response: As we mentioned in the previous submission,

*"The most recent efforts were made by Zou et al. (2017) using the mean annual temperature at the top of permafrost (TTOP) model (referenced as $QTP_{TTOP}$ map) forced by land surface temperature (or freezing and thawing indices) considering soil properties, and by Wu, Nan, Zhao, Zhao, Cheng, (2018) based on Noah land surface model (referenced as $QTP_{Noah}$ map) as well as gridded meteorological dataset (e.g., surface air temperature, radiation, and precipitation)"*

The land surface temperature used in the $QTP_{TTOP}$ map was calibrated based on ground observations or the station data, but only grid data was used by the $QTP_{Noah}$ map. We will change this sentence to

*"The most recent efforts were made by Zou et al. (2017) using the mean annual temperature at the top of permafrost (TTOP) model (referenced as $QTP_{TTOP}$ map) forced by **calibrated (using station data)** land surface temperature (or freezing and thawing indices) considering soil properties, and by Wu, Nan, Zhao, Zhao, Cheng, (2018) based on Noah land surface model (referenced as $QTP_{Noah}$ map) as well as gridded meteorological dataset, **including surface air temperature, radiation, and precipitation."***

to clarify.

- P3, L6: "(1)' –> "(i)"
  Response: Done.

- P3, L7: "(2)' –> "(ii)"
  Response: Done.

- P3, L11: "... the temperature at ..." –> "...the mean annual temperature at..."
  Response: Done.

- P4, L5: "... outline of QTP ..." –> "...outline of the QTP..."
  Response: Done.

- P4, L24: Is the calculation of "Cohen's kappa coefficient" too complicated? If not, please put equation(s) here and indicate what a high k means. Is there some threshold to roughly classify good, fair, or others?
  Response: "Cohen's kappa coefficient" equations will be added as

$$\kappa = \frac{p_o - p_e}{1 - p_e} \tag{1}$$

*"where $p_e$ and $p_o$ are the probability of random agreement and disagreement, respectively, can be calculated as"*

$$p_e = \frac{(PF_T + PF_F) \times (PF_T + NPF_F)}{(PF_T + PF_F + NPF_F + NPF_T)^2} \tag{2}$$

$$p_o = \frac{(NPF_F + NPF_T) \times (PF_F + NPF_T)}{(PF_T + PF_F + NPF_F + NPF_T)^2} \tag{3}$$

Authors will remove the kappa coefficient threshold description from footnote of Table 1 to this section.

*"Cohen's kappa coefficient result is interpreted as excellent agreement for $k \geqslant 0.8$, substantial agreement for $0.6 \leqslant k < 0.8$, moderate agreement for $0.4 \leqslant k < 0.6$, fair agreement for $0.2 \leqslant k < 0.4$, and slight agreement for $k < 0.2$."*

- P5, L12-13: What's Qxx?
  Response: Did not see "Qxx".

- P5, L15: Cao et al. (?), missing year.
  Response: It will be revised to "Cao et al. (2018)".

- P6, L14: why is "-3 to -4 °C"? Generally, -4 to -3 °C?
  Response: It will be revised to -4 to -3 °C.

**References**

Cao, B., Gruber, S., & Zhang, T. (2017). REDCAPP (v1.0): Parameterizing valley inversions in air temperature data downscaled from re-analyses. Geoscientific Model Development Discussions, 2017, 1–35. https://doi.org/10.5194/gmd-2017-60

Cao, B., Gruber, S., Zhang, T., Li, L., Peng, X., Wang, K., Zheng, L., Shao, W., Guo, H. (2017). Spatial variability of active layer thickness detected by ground-penetrating radar in the Qilian Mountains, Western China. Journal of Geophysical Research: Earth Surface. https://doi.org/10.1002/2016JF004018

Chen, Y., Yang, K., He, J., Qin, J., Shi, J., Du, J., & He, Q. (2011). Improving land surface temperature modeling for dry land of China. Journal of Geophysical Research Atmospheres, 116(20), 1–15. https://doi.org/10.1029/2011JD015921

Cremonese, E., Gruber, S., Phillips, M., Pogliotti, P., Boeckli, L., Noetzli, J., Noetzli, J., Suter, C., Bodin, X., Crepaz, A., Kellerer-Pirklbauer, A., Lang, -K., Letey, S., Mair, V., Morra di Cella, U., Ravanel, L., Scapozza, C., Seppi, R Kellerer-Pirklbauer, A. (2011). Brief Communication:An inventory of permafrost evidence for the European Alps. The Cryosphere, 5, 651–657.

Prince, S. D., Goetz, S. J., Dubayah, R. O., Czajkowski, K. P., & Thawley, M. (1998). Inference of surface and air temperature, atmospheric precipitable water and vapor pressure deficit

using advanced very high-resolution radiometer satellite observations: Comparison with field observations. Journal of Hydrology, 212–213(1–4), 230–249. https://doi.org/10.1016/S0022-1694(98)00210-8

Ran, Y., Li, X., Cheng, G., Zhang, T., Wu, Q., Jin, H., & Jin, R. (2012). Distribution of Permafrost in China: An Overview of Existing Permafrost Maps. Permafrost and Periglacial Processes, 23(4), 322–333. https://doi.org/10.1002/ppp.1756

Wu, X., Nan, Z., Zhao, S., Zhao, L., Cheng, G. (2018). Spatial modeling of permafrost distribution and properties on the Qinghai-Tibet Plateau. Permafrost and Periglacial Processes, 29(2), 86–99. https://doi.org/10.1002/ppp.1971

Yang, K., He, J., Tang, W., Qin, J., & Cheng, C. C. K. (2010). On downward shortwave and longwave radiations over high altitude regions: Observation and modeling in the Tibetan Plateau. Agricultural and Forest Meteorology, 150(1), 38–46. https://doi.org/https://doi.org/10.1016/j.agrformet.2009.08.004 Zou, D., Zhao, L., Sheng, Y., Chen, J., Hu, G., Wu, T., . . . Cheng, G. (2017). A new map of permafrost distribution on the Tibetan Plateau. The Cryosphere, 11(6), 2527–2542. https://doi.org/10.5194/tc-11-2527-2017

Zou, D., Zhao, L., Sheng, Y., Chen, J., Hu, G., Wu, T., Wu, J., Xie, C., Wu, X., Pang, Q., Wang, W., Du, E., Li, W., Liu, G., Li, J., Qin, Y., Qiao, Y., Wang, Z., Shi, J., Cheng, G. (2017). A new map of permafrost distribution on the Tibetan Plateau. The Cryosphere, 11(6), 2527–2542. https://doi.org/10.5194/tc-11-2527-2017

---

## Author Comment (AC2) · 31 Oct 2018

**Response to Anonymous Referee #2**

**Anonymous Referee #2**

The authors would like to thank the reviewer for the constructive feedback, and the thorough assessment of the manuscript. Below we provide a point-to-point response to each comment, reviewer comments are given in black, responses are given in blue. Additionally, we have included details of how we intend to address these changes in a revised submission.

The manuscript presents a useful contribution for understanding performance of different permafrost maps at QTP. The aim of the study, methods and presented results are relatively clear, however, several parts of the text need to be clarified and part of the methods needs to be slightly extended. The manuscript has to be proofread for language and use of several terms in the manuscript can be improved. I have listed a number of specific comments below, which should improve the clarity of the text. Response: The language of revised manuscript will be carefully checked.

Authors should find the comments straightforward to implement.

**Specific comments:**

- Page 1, line 4: change "overall accuracy of about" to "overall accuracy between"
  Response: Done.

- Page 1, line 5: omit "extremely large". The areas are matter of scale and don't need to be evaluated in this case. It is also not clear how this part of the sentence relates to the beginning where comparison to in-situ measurements is discussed.
  Response: Yes, they are compared in the manuscript rather than evaluated. In the revised manuscript, we will reformulate this part to

  *"Many maps have been produced to estimate permafrost distribution over the Qinghai-Tibet Plateau, however, the estimated permafrost region (1.42–1.84×10$^6$ km$^2$) and area (0.76–1.25×10$^6$ km$^2$) are extremely large. The evaluation and inter-comparisons of them are poorly understood due to limited evidence."*

- Page 1, line 6: How do you define "fragile landscapes"?
  Response: "fragile landscapes" means the areas where topography (mountains or valleys), surface conditions (e.g., vegetation cover, soil proxies, and river distribution) are spatial variable. The "fragile landscape" will be replaced by *"spatially highly variable landscape"* to clarify.

- Page 2, lines 4-5: What is a large enough dataset? I assume that the evaluation datasets were large enough for the publications to be published. In the next sentence, "This would weaken their applications" sounds as the datasets were inappropriate. I would change the formulations of the both sentence to more positive. For instance: "The new larger dataset can be used to improve evaluations of the existing datasets, which would further improve their applications..."

Response: In the revised manuscript, this part will be changed to

*"Despite the increasing efforts made on permafrost mapping, existing maps over the QTP so far have not been evaluated and inter-compared with large data sets. A large amount of permafrost presence/absence evidence has been collected using a wide variety of methods (e.g., ground temperature, soil pits, and geo-physics) on the QTP since the 2000s. The new larger dataset can be used to improve evaluations of the existing datasets, which would further improve their applications in permafrost and related studies, e.g., as a boundary condition for eco-hydrological model simulations."*

- Page 2, line 16: The word evidence is used at many places in the manuscript. I'm not sure that its use is correct. It could be replaced by "information" in this case and maybe just a "validation site" elsewhere in the manuscript.
Response: "Evidence" has been widely used for describing permafrost presence or absence "validation site". I listed several published literatures using "evidence" below.

*Cremonese, E., Gruber, S., Phillips, M., Pogliotti, P., Boeckli, L., Noetzli, J., . . . Zischg, A. (2011). Brief Communication: "An inventory of permafrost evidence for the European Alps." The Cryosphere, 5(3), 651–657. https://doi.org/10.5194/tc-5-651-2011*

*Boeckli, L., Brenning, A., Gruber, S., & Noetzli, J. (2012). A statistical approach to modelling permafrost distribution in the European Alps or similar mountain ranges. The Cryosphere, 6(1), 125–140. https://doi.org/10.5194/tc-6-125-2012*

*Schmid, M.-O., Baral, P., Gruber, S., Shahi, S., Shrestha, T., Stumm, D., &*

*Wester, P. (2015). Assessment of permafrost distribution maps in the Hindu Kush Himalayan region using rock glaciers mapped in Google Earth. The Cryosphere, 9(6), 2089–2099. https://doi.org/10.5194/tc-9-2089-2015*

We would keep the evidence in the revised manuscript, and hope you agree.

- Page 2, line 18: The use of word "confidence" shall be used instead of "certainty" also further in the manuscript.
  Response: Yes, we agree. In the revised manuscript, the "certainty" will be changed to "confidence".

- Page2, line 25: What are your criteria to define confidence (certainty) classes medium and low? How are these classes used further in the manuscript?
  Response: The confidence degree was described in the manuscript and available in the inventory as supplement, however, it was not further used for map evaluation. Since the inventory may be used for other related studies (e.g., permafrost simulation evaluation), and the confidence information would be useful for further selecting the data based on research aims, we would keep the classification in the inventory and move the classification description into the Appendix A (See below).

The classification algorithm of confidence degree largely follows Cremonese et al. (2011) and could be summarized as

*"For board use of the permafrost presence or absence inventory, the data confidence degree was provided (Table A1). BH and SP provide direct evidence of permafrost presence or absence based on MAGT and/or ground ice observations, and hence have high confidence (Cremonese et al., 2011). The data confidence derived from MAGST is classified based on temperature and the length*

**Table *A1.*** *Classification algorithm of in-situ permafrost presence or absence evidence from various methods*

| Method | Indicator | Survey depth | Permafrost | Confidence degree |
|---|---|---|---|---|
| BH | $MAGT \leqslant 0\ °C$ | meters to about 20 m | presence | high |
| SP | ground ice presence | about 1.0–2.5 m | presence | high |
| GST | $MAGST \leqslant \text{-}2\ °C\ \&\ observations \geqslant 3$ | 0.05 or 0.1 m | presence | medium |
| | $MAGST \leqslant \text{-}2\ °C\ \&\ observations < 3$ | | presence | low |
| | $MAGST > \text{-}2\ °C\ \&\ MAGST + TO_{max} \leqslant 0\ °C$ | | presence | low |
| | $MAGST < 0\ °C\ \&\ MAGST + TO_{max} > 0\ °C$ | | ambiguous | – |
| | $MAGST > 0\ °C$ | | absence | medium |
| GPR | clear permafrost reflection | about 0.80–5.0 m | presence | medium |

*BH = borehole temperature, SP = soil pit, GST = ground surface temperature, and GPR = ground-penetrating radar. $TO_{max}$, the maximum thermal offset under natural conditions reported for the QTP, is 0.79 °C.*

of the observation period. The evaluated GPR survey result was considered as medium confidence."

- Page 3, lines 4-5: How do you define a clear permafrost reflection? The exact criteria for selection of GPR sites should be presented.
Response: Cao et al. (2017) presented detailed description of GPR data acquisition and processing, here we used the data which active layer depth was identified, and could summarized as

*"Here, GPR data from Cao et al. (2017) are measured using 100 and 200 MHz antennas depending on the active layer thickness. The GPR survey depth is from about 0.8 to near 5 m, and the data are considered as indicating the presence of permafrost only if an active layer thickness (or a clear permafrost reflection) could be established."*

to clarify.

- Page 3, line 9: The IPA map shows extent of four permafrost zones and is therefore not a binary map. Present here how did you convert it in to binary map showing permafrost presence and absence.
Response: Yes, the IPA map is categorical map rather than binary. Additionally, the $QTP_{TTOP}$ and $QTP_{Noah}$ maps are also categorical maps. The binary map was changed to categorical map throughout the manuscript. In the revised manuscript, we will change this part to

*"In general, permafrost maps over the QTP could be classified as (i) categorical, using categorical classification with different permafrost types (e.g., continuous, discontinuous, sporadic, and island permafrost), seasonally frozen ground, and unfrozen ground, and (ii) continuous, using continuous probability or indices [0–1] to represent proportion of an area that is underlain by permafrost."*

to clarify.

In Section 2.4, we will also add

*"For map evaluation, the categorical map was aggregated to binary map by merging different permafrost types to permafrost presence [1] and by merging the others to permafrost absence [0]."*

- Page 3, line 16: Please explain here how PZIcold, PZIwarm and PZInorm were derived by Gruber (2012) and what is difference between them.
Response: As we mentioned in the previous manuscript, the $PZI_{global}$ map is derived largely based on the heuristic-empirical relationship between PZI and mean annual air temperature (MAAT) based on generalized linear models. The model parameters are established largely based on the boundaries of continuous (PZI = 0.9 for MAAT = -8.0 °C) and isolated (PZI = 0.1 for MAAT = -1.5 °C) permafrost in the IPA map and do not use field observations. The cold and warm cases were introduced into the map to allow the propagation of uncertainty caused by input dataset and model suitability, and they differ in the parameters used. Comparing the normal case, the cold and warm variants are derived by shifting PZI and MAAT at the respective limit by $\pm$ 5% and $\pm$ 0.5 °C, respectively. We will change this part to

*"The model parameters are established largely based on the boundaries of continuous (PZI = 0.9 for MAAT = -8.0 °C) and isolated (PZI = 0.1 for MAAT = -1.5 °C) permafrost in the IPA map and do not use field observations. Additionally, two cases, including cold (conservative or more permafrost) and warm (anti-conservative or less permafrost), were introduced into the map to allow the propagation of uncertainty caused by input dataset and model suitability. The three cases, and hence the $PZI_{norm}$, $PZI_{warm}$, and $PZI_{cold}$ maps, differ in the*

*parameters used. Comparing the normal case, the cold and warm variants are derived by shifting PZI and MAAT at the respective limit by ± 5% and ± 0.5 °C, respectively."*

- Page 3, consider moving 2.3 section before 2.2 because it is in my opinion logical continuation of the inventory of permafrost validation sites. Also consider changing the section title to "Topographical and climatological properties of the inventory (or permafrost validation) sites"
  Response: In the revised manuscript, the section 2.3 will be moved before 2.2, and the title will be changed to
  *"Topographical and climatological properties of the inventory sites".*

- Page 3, line 32: What are you referring to with "(about 500m)"?
  Response: It is the spatial resolution. In the revised manuscript, the sentence will be changed to
  *"The MASCD with a spatial resolution of about 500 m was. . ."*

- Page 4, line 9: Please consider extending the explanation about the difference between permafrost area and permafrost region. This concept is difficult to understand by broader permafrost community. Maybe introduce the concept of scale and ground coverage by permafrost.
  Response: This part will be changed to

  *"Permafrost region is the exposed land surface below which permafrost is likely present, however, the permafrost may not be everywhere. It is an arbitrary definition, and usually 10% permafrost coverage is used as the threshold (Gruber , 2012). Permafrost area is where actually underlain by permafrost (Zhang et al., 2000)."*

[Figure]

- Page 4, lines 26-27: Restructure the sentences. It sounds as because of your permafrost absence/absence classification, you have 1475 sites. I assume that this is because of your site selection criteria.
Response: In the revised manuscript, the sentence will be change to

  *"In the inventory, there are in total 1475 permafrost presence or absence sites/plots acquired from BH, SP, GST, and GPR methods (Figure 1)."*

- Page 5, line 3: "were aggregated based on their major value". Maybe replace with "the majority value was assigned to aggregated sites.
Response: Done.

- Page 5, line 15: More appropriate term for "band" would be "range". What exactly does the word "sensitive" refer to?
Response: "band" will be replaced by "range" throughout the revised manuscript. The sentence will be deleted as it does not give us too much useful information.

- Page 5, line 31: Did you mean $QTP_{TTOP}$ instead of $PZI_{TTOP}$?
Response: Yes, it will be corrected.

- Page6, line 10: Again, how exactly is fragile landscape defined?
Response: The "fragile landscape" will be replaced by *"spatially highly variable landscape"*.

**References**

Cao, B., Gruber, S., Zhang, T., Li, L., Peng, X., Wang, K., Zheng, L., Shao, W., Guo, H. (2017). Spatial variability of active layer thickness detected by ground-penetrating radar in the Qilian Mountains, Western China. Journal of Geophysical Research: Earth Surface.

https://doi.org/10.1002/2016JF004018

Cremonese, E., Gruber, S., Phillips, M., Pogliotti, P., Boeckli, L., Noetzli, J., Noetzli, J., Suter, C., Bodin, X., Crepaz, A., Kellerer-Pirklbauer, A., Lang, -K., Letey, S., Mair, V., Morra di Cella, U., Ravanel, L., Scapozza, C., Seppi, R Kellerer-Pirklbauer, A. (2011). Brief Communication:An inventory of permafrost evidence for the European Alps. The Cryosphere, 5, 651–657.

Gruber, S.: Derivation and analysis of a high-resolution estimate of global permafrost zonation, The Cryosphere, 6, 221–233, https://doi.org/10.5194/tc-6-221-2012, 2012.

Zhang, T., Heginbottom, J. A., Barry, R. G., and Brown, J.: Further statistics on the distribution of permafrost and ground ice in the Northern Hemisphere, Polar Geography, 24, 126–131, https://doi.org/ 10.1080/10889370009377692, 2000.

---

## Author Response (AR1)

**Author's Response on *"Brief Communication:Evaluation and comparisons of permafrost map over Qinghai-Tibet Plateau based on inventory of in-situ evidence"**

Bin Cao[1,2], Tingjun Zhang[1], Qingbai Wu[3], Yu Sheng[3], Lin Zhao[4], and Defu Zou[4]

[1]Key Laboratory of Western China's Environmental Systems (Ministry of Education), College of Earth and Environmental Sciences, Lanzhou University, Lanzhou 730000, China
[2]Department of Geography & Environmental Studies, Carleton University, Ottawa K1S 5B6, Canada
[3]State Key Laboratory of Frozen Soil Engineering, Cold and Arid Regions Environmental and Engineering Research Institute, Chinese Academy of Sciences, Lanzhou 730000, China
[4]Cryosphere Research Station on the Qinghai-Tibet Plateau, State Key Laboratory of Cryospheric Science, Cold and Arid Regions Environmental and Engineering Research Institute, Chinese Academy of Sciences, Lanzhou 730000, China

**Correspondence**: Tingjun Zhang(tjzhang@lzu.edu.cn)

The authors would like to thank Peter Morse and two anonymous reviewers for the constructive feedback, and the thorough assessment of the manuscript. Below we provide a point-to-point response to each comment, reviewer comments are given in black, responses are given in blue. Additionally, we have included details of how we addressed these changes in the revised submission.

**Response to Anonymous Referee #1**

Permafrost maps were released by various institutes or research teams during the past several decades. They used modeling, statistical, and other mapping techs. Basically, the maps were evaluated during processing. However, the inter-comparison, what this study was done, is required for better understanding. This study collected more than a thousand samples over the QTP. The results of this study would be useful for future permafrost studies on the QTP and broad interest to the permafrost communities.

The manuscript, however, requires a bit more work before it is acceptable for publication. For the most part, the manuscript is well written but some editing is required to improve language and increase clarity. There are a few places in the manuscript where more explanation would be helpful.
Response: The language of revised manuscript was carefully checked.

Although I have made a few comments here that I hope the authors will find useful, dealing with them may not take too much time. The authors should thoroughly proofread the revised manuscript before submission or invite a native speaker in permafrost communities to improve the language. I am willing to review the revised paper.

**Major:**

- **Unclear description and logic (to the following results) in the Data and Methods section.**
  The authors used four methods to classify permafrost or not. However, it's not enough for understanding, although this paper is a short communication.

  – How deep are generally for boreholes and soil pits? 1 m, 5 m?
  Response: In general, the borehole depths vary from meters to hundred meters. In this study, we used the mean annual ground temperature from boreholes, which also varies from several meters to about 20 m, to identify permafrost presence. Number of samples measured from soil pits was small (6 samples) due to the prevalent coarse soil, and their depths are between less than 1 m to about 2.5 m. We added

  *"In this study, we used the mean annual ground temperature (MAGT) measured from boreholes, which varies from meters to about 20 m to identify permafrost presence or absence. Due to the prevalent coarse soil, SP was only applied in areas possible, and the depth is from less than 1 meter to about 2.5 m."*

  to clarify.

  Additionally, the survey depth of all the methods is summarized in Table A1 (see below)

  – It looks like this study used only $MAGST + TO_{max} \leqslant 0$ as the standard. In your results, you only talked about the sites considered as permafrost. I am not sure whether these classifications (P2, L25-29) are necessary.
  Response: As mentioned by Referee #2 (specific comment Page 2, line 18), the word "certainty" is changed to "confidence". Yes, the confidence classifications were not further used in this manuscript, but only present in the inventory as supplement. Since the inventory may be useful for other researches, we would keep the classification in the inventory and move the classification description into the Appendix A (see below). We hope you agree.

  The classification algorithm of confidence degree largely follows Cremonese et al. (2011) and could be summarized as

  *"Appendix A: Classification algorithm of in-situ permafrost presence or absence evidence*

  *For board use of the permafrost presence or absence inventory, the data confidence degree was provided (Table A1). BH and SP provide direct evidence of permafrost presence or absence based on MAGT and/or ground ice observations, and hence have high confidence (Cremonese et al., 2011). The data confidence derived from MAGST is classified based on temperature and the length of the observation period. The evaluated GPR survey result was considered as medium confidence."*

*Table A1*: *Classification algorithm of in-situ permafrost presence or absence evidence from various methods*

| Method | Indicator | Survey depth | Permafrost | Confidence degree |
|---|---|---|---|---|
| BH | $MAGT \leqslant 0$ | meters to about 20 m | presence | high |
| SP | ground ice presence | about 1.0–2.5 m | presence | high |
| GST | $MAGST \leqslant -2\ °C$ & observations $\geqslant 3$ | 0.05 or 0.1 m | presence | medium |
| | $MAGST \leqslant -2\ °C$ & observations $< 3$ | | presence | low |
| | $MAGST > -2\ °C$ & $MAGST + TO_{max} \leqslant 0\ °C$ | | presence | low |
| | $MAGST < 0\ °C$ & $MAGST + TO_{max} > 0\ °C$ | | ambiguous | – |
| | $MAGST > 0\ °C$ | | absence | medium |
| GPR | clear permafrost reflection | about 0.80–5.0 m | presence | medium |

BH = borehole temperature, SP = soil pit, GST = ground surface temperature, and GPR = ground-penetrating radar.
$TO_{max}$, the maximum thermal offset under natural conditions reported for the QTP, is 0.79 °C.

– What is kind of antennas generally used in GPR survey? Also, how deep is accessed?
Response: The GPR survey was conducted using 100 and 200 MHz antennas and evaluated using direct measurements (e.g., mechanical probing, soil temperature, and soil pits) (Cao et al. , 2017). The survey depth was from about 0.8 to near 5 m depending on the active layer thickness. Authors added

*"Here, GPR data from Cao et al. (2017) are measured using 100 and 200 MHz antennas depending on the active layer thickness. The GPR survey depth is from about 0.8 to near 5 m, and the data are considered as indicating the presence of permafrost only if an active layer thickness (or a clear permafrost reflection) could be established."*

to clarify.

– In section 2.3, you used DEM (3 arc second), MAAT (1 km), MASCD (∼500 m), and NDVI (∼250 m). I guess you extract those variables for each site in your inventory. Is it?
Response: First of all, it is moved to Section 2 as Referee #2 suggested. Yes. We extract these variables to sample sites using nearest interpolation. We added

*"These climate variables were extracted to in-situ sites/plots based on nearest interpolation."*

to clarify.

You also said (P5, L2-4) "Where original field evidence of permafrost presence/absence is located within the same grid cell (30 arcsec, 1 km), they were aggregated based on their major value. For a grid with one permafrost site and one non-permafrost site, the nearer site from the grid center was used to represent the grid." (actually, these sentences should be moved to section 2.3). Why did you have to aggregate these in-situ data to 1 km?
Response: Yes, they are located within the same grid cell of unprojected SRTM30 with a spatial resolution of 30 arcsec. The aggregation was deleted in the revised manuscript.

How did you deal with DEM, MASCD, and NDVI? Did you upscale DEM, MASCD, and NDVI to 1 km? I guess you were going to avoid conflict sites (permafrost and non-permafrost) in the same pixel. Is it?
Response: No. The 3 arcsec DEM was used to simulate the slope and aspect for the in-situ sites. The MASCD,

NDVI, and MAAT are used here to explore the representive of the inventory, and they are extracted to the sites based on nearest interpolation.

When you extracted values from different spatial resolution datasets, even if there are probably few sites in the same pixel at 1 km resolution, however, there still are three spatial datasets with higher resolution, which might bring different snow, topography, and vegetation condition to your sites. In fact, there might be different ground thermal states under the same climate and vegetation condition because of different soil wetness, soil properties, and so on. Overall, I don't think the aggregation is necessary.

Response: Yes, we agree. We omitted the evidence aggregation and conduct the evaluation with all the 1475 sites. Please note that, some statistics may be slightly different by using the original 1475 evaluation sites/plots.

Furthermore, how did you compare with the maps with different spatial scale, e.g., QTP$_{Noah}$ map is 10 km. Those issues were confusing and should be clarified.

Response: The evaluation was conducted at the sites we collected. Permafrost presence and absence information at evaluation sites was extracted to the evidence based on nearest from different maps. In Section 2.4 Statistics and evaluation of permafrost distribution maps, we added

*"The permafrost and absence information was extracted to in-situ sites, and..."*

to clarify.

- Misleading indicators.

PCC$_{PF}$, PCC$_{NPF}$, and PCC$_{tol}$ were used to quantify the classification accuracies of permafrost maps. To my sense, PCC$_{PF}$ and PCC$_{NPF}$ are not useful and may be misleading. When the map over-presents permafrost (i.e., much colder), PCC$_{PF}$ would be extremely close to 100%. Can we say this is much better? Vice versa. Thus, the description in Section 3.2 could be misleading, at least to me, and should be more cautious. I suggest removing those parts.

Response: Yes, we agree that PCC$_{PF}$ and PCC$_{NPF}$ are somehow misleading when we look at them separately without due care. On the other hand, these two indictors would be useful if they are jointly interpolated. As you mentioned, the high PCC$_{PF}$ together with low PCC$_{NPF}$ indicate the map over-presents permafrost. This information could not be indicated by either kappa coefficient nor PCC$_{tol}$. For this reason, we would keep these three indicators. To reduce the misunderstanding, the PCC$_{PF}$ and PCC$_{NPF}$ are interpolated together throughout the manuscript, and the over- or less-presents permafrost was also present. Additionally, the PCC$_{tol}$ in Figure 2 was deleted to avoid confusion. In Section 3.2 Evaluation and comparison of existing maps, we added

*"The high PCC$_{PF}$ together with low PCC$_{NPF}$ for the IPA, QTP$_{Noah}$, PZI$_{cold}$, and QTP$_{TTOP}$ maps indicate permafrost is over-presented by them, while the PZI$_{warm}$ and PZI$_{norm}$ maps showed underestimated the permafrost over the QTP."*

Meanwhile, do you consider the effect of the different sample volume? Because in your in-situ sites pool, number of sites with permafrost is twice as large as the sites without permafrost.

Response: Yes, the kappa coefficient, *"which measures inter-rater agreement for categorical items"*, was introduced here as the major indicator for map evaluation as it could largely *"avoid the impact of uneven distribution of sample numbers for permafrost presence and absence"*.

- More discussion?

  This study found different performance in permafrost maps. It's better to discuss a little bit more about the sources of bias, such as different MAAT products. More discussion on the possible sources of the revealed differences would enhance the scientific significance. Meanwhile, it also is useful for the future permafrost map updating.

  Response: Yes, we agree. Our previous manuscript had partly discussed the bias from inputs for the $QTP_{Noah}$ and IPA maps. We enhanced this part, and the inputs bias was discussed for each map as below:

  $QTP_{Noah}$ *map: "Though the $QTP_{Noah}$ map was derived using coupled land surface model (Noah), the relatively worse performance, especially for non-permafrost area ($PCC_{NPF}$ = 45.9%), is likely caused by inputting coarse-scale forcing dataset (0.1° resolution or ~10 km) (Chen et al., 2011) and by the uncertainty of soil texture dataset (Yang et al., 2010)."*

  *IPA map:"It is not surprising that the IPA map has fair agreement (k = 0.32) as less observations were compiled and the method used are more suitable for high latitudes (Ran et al., 2012)."*

  $QTP_{TTOP}$ *map: "The $QTP_{TTOP}$ map was derived based on MODIS land surface temperature with different temporal coverage of 2003–2012 (Zou et al., 2017). Though the MODIS land surface temperature time-series gaps caused mainly by cloud were filled using the Harmonic Analysis Time-Series (HANTS) algorithm (Prince et al., 1998), the surface conditions, especially vegetation and snow cover, were ignored. In this case, land surface temperature is underestimated in high and/or dense vegetation area as it comes from the top of vegetation canopy, and is overestimated in snow covered area due to the cooling effects of snow is not considered. As a consequence, permafrost is likely overestimated in high and/or dense vegetation area and underestimated in regular snow-covered area."*

  $PZI_{global}$ *map: "The MAAT used in the $PZI_{global}$ map was statistical downscaled based on the lapse rate from the upper-air (or pressure level) temperature of NCEP, but the influences of land surface on surface air temperature, such as cold air pooling, was ignored (Cao et al., 2017). This is important as winter inversion is excepted to be common due to the prevalent mountains over the QTP. In other words, permafrost may be underestimated in valleys due to the overestimated MAAT."*

**Specific:**

- P1, title: the title could be "Ground-based evaluation and inter-comparisons of permafrost maps over the Qinghai-Tibet Plateau"?

  Response: We changed the title to

  *"Evaluation and inter-comparisons of permafrost map over the Qinghai-Tibet Plateau based on inventory of in-situ evidence".*

  As the study also provided the first inventory of permafrost presence or absence over the Qinghai-Tibet Plateau based on in-situ evidence, authors would like to reflect the inventory in the title. I hope you agree.

- P1, L3: the number, 1475, might be misleading although you collected. Because you aggregated to 1040, which excluded about 400 sites. Add a comma to 1040/1475 for consistency.

  Response: The aggregation part was removed, and evaluation was conducted using all the data.

- P2, L1: "hemisphere" –> hemispheric ?
  Response: Corrected.

- P2, L10: "2000" –> "the 2000s"?
  Response: Corrected.

- P2, L16: insert "survey" after GPR.
  Response: Done.

- P2, L25-29: Where is so-call "high certainty" for permafrost classification? Meanwhile, it looks like this study used only $MAGST + TO_{max} \leqslant 0$ as the standard. I am not sure whether these classifications are necessary. If necessary, the authors should clarify.
  Response: As the Referee #2 mentioned, the word "certainty" is changed to "confidence". The evidence derived from BH and SP is considered as high confidence as they provide direct information, such as mean annual ground temperature or ground ice presence. Yes. To determine permafrost presence or absence, only the function of $MAGST + TO_{max} \leqslant 0$ °C was used. The confidence classifications were not used in this manuscript, but only present in the inventory as supplement. Since the inventory may be used for other related studies (e.g., permafrost simulation evaluation), and the confidence information would be useful for further selecting the data, we would keep the classification in the inventory and move the classification description into the Appendix A. Please also see our response to the major comments of "Unclear description and logic".

- P3, L1: The authors should briefly clarify what kind of antennas were used and how deep is accessible.
  Response: The author added

  *"Here, GPR data from Cao et al. (2017) are measured using 100 and 200 MHz antennas depending on the active layer thickness. The GPR survey depth is from about 0.8 to near 5 m, and the data are considered as indicating the presence of permafrost only if only an active layer thickness (or a clear permafrost reflection) could be established."*

  to clarify.

- P3, Section 2.2: It's worth to note what climate data were used in QTP$_{TTOP}$ and QTP$_{Noah}$ maps. Both used the data merged MODIS temperature products and station data?
  Response: As we mentioned in the previous submission,

  *"The most recent efforts were made by Zou et al. (2017) using the mean annual temperature at the top of permafrost (TTOP) model (referenced as QTP$_{TTOP}$ map) forced by land surface temperature (or freezing and thawing indices) considering soil properties, and by Wu et al. (2018) based on Noah land surface model (referenced as QTP$_{Noah}$ map) as well as gridded meteorological dataset (e.g., surface air temperature, radiation, and precipitation)"*

  The land surface temperature used in the QTP$_{TTOP}$ map was calibrated based on ground observations or the station data, but only grid data was used by the QTP$_{Noah}$ map. We changed this sentence to

  *"The most recent efforts were made by Zou et al. (2017) using the mean annual temperature at the top of permafrost (TTOP) model (referenced as QTP$_{TTOP}$ map) forced by **calibrated (using station data)** land surface temperature (or freezing and thawing indices) considering soil properties, and by Wu et al. (2018) based on*

*Noah land surface model (referenced as $QTP_{Noah}$ map) as well as gridded meteorological dataset, **including surface air temperature, radiation, and precipitation.”***

to clarify.

- P3, L6: “(1)’ –> “(i)”
  Response: Done.

- P3, L7: “(2)’ –> “(ii)”
  Response: Done.

- P3, L11: “... the temperature at ...” –> “...the mean annual temperature at...”
  Response: Done.

- P4, L5: “... outline of QTP ...” –> “...outline of the QTP...”
  Response: Done.

- P4, L24: Is the calculation of “Cohen’s kappa coefficient” too complicated? If not, please put equation(s) here and indicate what a high k means. Is there some threshold to roughly classify good, fair, or others?
  Response: “Cohen’s kappa coefficient” equations were added as

$$\kappa = \frac{p_o - p_e}{1 - p_e} \tag{1}$$

where $p_e$ and $p_o$ are the probability of random agreement and disagreement, respectively, can be calculated as

$$p_e = \frac{(PF_T + PF_F) \times (PF_T + NPF_F)}{(PF_T + PF_F + NPF_F + NPF_T)^2} \tag{2}$$

$$p_o = \frac{(NPF_F + NPF_T) \times (PF_F + NPF_T)}{(PF_T + PF_F + NPF_F + NPF_T)^2} \tag{3}$$

Authors removed the kappa coefficient threshold description from footnote of Table 1 to this section (see below).

*”Cohen’s kappa coefficient result is interpreted as excellent agreement for $k \geqslant 0.8$, substantial agreement for $0.6 \leqslant k < 0.8$, moderate agreement for $0.4 \leqslant k < 0.6$, fair agreement for $0.2 \leqslant k < 0.4$, and slight agreement for $k < 0.2$.”*

- P5, L12-13: What’s Qxx?
  Response: Did not see “Qxx”.

- P5, L15: Cao et al. (?), missing year.
  Response: It was revised to *“Cao et al. (2018)”*.

- P6, L14: why is “-3 to -4 C”? Generally, -4 to -3 C?
  Response: It was revised to -4 to -3 °C.

**Response to Anonymous Referee #2**

The manuscript presents a useful contribution for understanding performance of different permafrost maps at QTP. The aim of the study, methods and presented results are relatively clear, however, several parts of the text need to be clarified and part of the methods needs to be slightly extended. The manuscript has to be proofread for language and use of several terms in the manuscript can be improved. I have listed a number of specific comments below, which should improve the clarity of the text.

Response: The language of revised manuscript was carefully checked.

Authors should find the comments straightforward to implement.

**Specific comments:**

- Page 1, line 4: change "overall accuracy of about" to "overall accuracy between"
  Response: Done.

- Page 1, line 5: omit "extremely large". The areas are matter of scale and don't need to be evaluated in this case. It is also not clear how this part of the sentence relates to the beginning where comparison to in-situ measurements is discussed.
  Response: Yes, they are compared in the manuscript rather than evaluated. We reformulated this part to

  *"Many maps have been produced to estimate permafrost distribution over the Qinghai-Tibet Plateau, however, the estimated permafrost region (1.42–1.84×10$^6$ km$^2$) and area (0.76–1.25×10$^6$ km$^2$) are extremely large. The evaluation and inter-comparisons of them are poorly understood due to limited evidence."*

- Page 1, line 6: How do you define "fragile landscapes"?
  Response: "fragile landscapes" means the areas where topography (mountains or valleys), surface conditions (e.g., vegetation cover, soil proxies, and river distribution) are spatially highly variable. The "fragile landscape" was replaced by *"spatially highly variable landscape"* to clarify.

- Page 2, lines 4-5: What is a large enough dataset? I assume that the evaluation datasets were large enough for the publications to be published. In the next sentence, "This would weaken their applications" sounds as the datasets were inappropriate. I would change the formulations of the both sentence to more positive. For instance: "The new larger dataset can be used to improve evaluations of the existing datasets, which would further improve their applications..."
  Response: This part was changed to

  *"Despite the increasing efforts made on permafrost mapping, existing maps over the QTP so far have not been evaluated and inter-compared with large data sets. A large amount of permafrost presence/absence evidence has been collected using a wide variety of methods (e.g., ground temperature, soil pits, and geophysics) on the QTP since the 2000s. The new larger dataset can be used to improve evaluations of the existing datasets, which would further improve their applications in permafrost and related studies, e.g., as a boundary condition for eco-hydrological model simulations."*

- Page 2, line 16: The word evidence is used at many places in the manuscript. I'm not sure that its use is correct. It could be replaced by "information" in this case and maybe just a "validation site" elsewhere in the manuscript.
  Response: "Evidence" has been widely used for describing permafrost presence or absence "validation site". I listed several published literatures using "evidence" below.

*Cremonese, E., Gruber, S., Phillips, M., Pogliotti, P., Boeckli, L., Noetzli, J., Noetzli, J., Suter, C., Bodin, X., Crepaz, A., Kellerer-Pirklbauer, A., Lang, K., Letey, S., Mair, V., Morra di Cella, U., Ravanel, L., Scapozza, C., Seppi, R Kellerer-Pirklbauer, A. (2011). Brief Communication: "An inventory of permafrost evidence for the European Alps." The Cryosphere, 5(3), 651–657. https://doi.org/10.5194/tc-5-651-2011*

*Boeckli, L., Brenning, A., Gruber, S., & Noetzli, J. (2012). A statistical approach to modelling permafrost distribution in the European Alps or similar mountain ranges. The Cryosphere, 6(1), 125–140. https://doi.org/10.5194/tc-6-125-2012*

*Schmid, M.-O., Baral, P., Gruber, S., Shahi, S., Shrestha, T., Stumm, D., & Wester, P. (2015). Assessment of permafrost distribution maps in the Hindu Kush Himalayan region using rock glaciers mapped in Google Earth. The Cryosphere, 9(6), 2089–2099. https://doi.org/10.5194/tc-9-2089-2015*

We would keep "evidence" in the revised manuscript, and hope you agree.

- Page 2, line 18: The use of word "confidence" shall be used instead of "certainty" also further in the manuscript.
Response: Yes, we agree. The "certainty" was changed to "confidence".

- Page2, line 25: What are your criteria to define confidence (certainty) classes medium and low? How are these classes used further in the manuscript?
Response: The confidence degree was described in the manuscript and available in the inventory as supplement, however, it was not further used for map evaluation. Since the inventory may be used for other related studies (e.g., permafrost simulation evaluation), and the confidence information would be useful for further selecting the data based on research aims, we would keep the classification in the inventory and move the classification description into the Appendix A (See below).

The classification algorithm of confidence degree largely follows Cremonese et al. (2011) and could be summarized as

***"Appendix A: Classification algorithm of in-situ permafrost presence or absence evidence***

*"For board use of the permafrost presence or absence inventory, the data confidence degree was provided (Table A1). BH and SP provide direct evidence of permafrost presence or absence based on MAGT and/or ground ice observations, and hence have high confidence (Cremonese et al., 2011). The data confidence derived from MAGST is classified based on temperature and the length of the observation period. The evaluated GPR survey result was considered as medium confidence.*

*Table A1*: *Classification algorithm of in-situ permafrost presence or absence evidence from various methods*

| Method | Indicator | Survey depth | Permafrost | Confidence degree |
|--------|-----------|--------------|------------|-------------------|
| BH | $MAGT \leqslant 0$ | meters to about 20 m | presence | high |
| SP | ground ice presence | about 1.0–2.5 m | presence | high |
| GST | $MAGST \leqslant -2\ °C$ & observations $\geqslant 3$ | 0.05 or 0.1 m | presence | medium |
| | $MAGST \leqslant -2\ °C$ & observations $< 3$ | | presence | low |
| | $MAGST > -2\ °C$ & $MAGST + TO_{max} \leqslant 0\ °C$ | | presence | low |
| | $MAGST < 0\ °C$ & $MAGST + TO_{max} > 0\ °C$ | | ambiguous | – |
| | $MAGST > 0\ °C$ | | absence | medium |
| GPR | clear permafrost reflection | about 0.80–5.0 m | presence | medium |

*BH = borehole temperature, SP = soil pit, GST = ground surface temperature, and GPR = ground-penetrating radar.*
*$TO_{max}$, the maximum thermal offset under natural conditions reported for the QTP, is 0.79 °C.*

- Page 3, lines 4-5: How do you define a clear permafrost reflection? The exact criteria for selection of GPR sites should be presented.
  Response: Cao et al. (2017) presented detailed description of GPR data acquisition and processing, here we used the data which active layer depth was identified, and could summarized as

  *"Here, GPR data from Cao et al. (2017) are measured using 100 and 200 MHz antennas depending on the active layer thickness. The GPR survey depth is from about 0.8 to near 5 m, and the data are considered as indicating the presence of permafrost only if an active layer thickness (or a clear permafrost reflection) could be established."*

  to clarify.

- Page 3, line 9: The IPA map shows extent of four permafrost zones and is therefore not a binary map. Present here how did you convert it in to binary map showing permafrost presence and absence.
  Response: Yes, the IPA map is categorical map rather than binary. Additionally, the QTP$_{TTOP}$ and QTP$_{Noah}$ maps are also categorical maps. The binary map was changed to categorical map throughout the manuscript. We changed this part to

  *"In general, permafrost maps over the QTP could be classified as (i) categorical, using categorical classification with different permafrost types (e.g., continuous, discontinuous, sporadic, and island permafrost), seasonally frozen ground, and unfrozen ground, and (ii) continuous, using continuous probability or indices [0–1] to represent proportion of an area that is underlain by permafrost."*

  to clarify.

  In Section 2.4, we also added

  *"For map evaluation, the categorical map was aggregated to binary map by merging different permafrost types to permafrost presence [1] and by merging the others to permafrost absence [0]."*

- Page 3, line 16: Please explain here how PZIcold, PZIwarm and PZInorm were derived by Gruber (2012) and what is difference between them.
  Response: As we mentioned in the previous manuscript, the PZI$_{global}$ map is derived largely based on the

heuristic-empirical relationship between PZI and mean annual air temperature (MAAT) based on generalized linear models. The model parameters are established largely based on the boundaries of continuous (PZI = 0.9 for MAAT = -8.0 °C) and isolated (PZI = 0.1 for MAAT = -1.5 °C) permafrost in the IPA map and do not use field observations. The cold and warm cases were introduced into the map to allow the propagation of uncertainty caused by input dataset and model suitability, and they differ in the parameters used. Comparing the normal case, the cold and warm variants are derived by shifting PZI and MAAT at the respective limit by ± 5% and ± 0.5 °C, respectively. We changed this part to

*"The model parameters are established largely based on the boundaries of continuous (PZI = 0.9 for MAAT = -8.0 °C) and isolated (PZI = 0.1 for MAAT = -1.5 °C) permafrost in the IPA map and do not use field observations. Additionally, two cases, including cold (conservative or more permafrost) and warm (anti-conservative or less permafrost), were introduced into the map to allow the propagation of uncertainty caused by input dataset and model suitability. The three cases or maps, referenced as $PZI_{norm}$, $PZI_{warm}$, and $PZI_{cold}$ maps, differ in the parameters used. Comparing the normal case, the cold and warm variants are derived by shifting PZI and MAAT at the respective limit by ± 5% and ± 0.5 °C, respectively."*

- Page 3, consider moving 2.3 section before 2.2 because it is in my opinion logical continuation of the inventory of permafrost validation sites. Also consider changing the section title to "Topographical and climatological properties of the inventory (or permafrost validation) sites"
  Response: The section 2.3 was moved before 2.2, and the title was changed to

  *"Topographical and climatological properties of the inventory sites".*

- Page 3, line 32: What are you referring to with "(about 500m)"?
  Response: It is the spatial resolution. The sentence was changed to

  *"The MASCD with a spatial resolution of about 500 m was. . . "*

- Page 4, line 9: Please consider extending the explanation about the difference between permafrost area and permafrost region. This concept is difficult to understand by broader permafrost community. Maybe introduce the concept of scale and ground coverage by permafrost.
  Response: This part was changed to

  *"Permafrost region refers to regions where permafrost exists but the entire region is not necessarily completely occupied by permafrost, while permafrost area refers to areas where are completely underlain by permafrost. For example, discontinuous permafrost regions have permafrost area ranging from 50 to 90%. In other words, in discontinuous permafrost region, there 50 to 90% of the area is underlain by permafrost, i.e., permafrost area (Zhang., 2000; Gruber , 2012)."*

- Page 4, lines 26-27: Restructure the sentences. It sounds as because of your permafrost absence/absence classification, you have 1475 sites. I assume that this is because of your site selection criteria.
  Response: The sentence was changed to

  *"In the inventory, there are in total 1475 permafrost presence or absence sites/plots acquired from BH, SP, GST, and GPR methods (Figure 1)."*

- Page 5, line 3: "were aggregated based on their major value". Maybe replace with "the majority value was assigned to aggregated sites.
  Response: Done.

- Page 5, line 15: More appropriate term for "band" would be "range". What exactly does the word "sensitive" refer to?
  Response: "band" was replaced by "range" throughout the revised manuscript. The sentence was deleted as it does not give us too much useful information.

- Page 5, line 31: Did you mean QTP$_{TTOP}$ instead of PZI$_{TTOP}$?
  Response: Yes, it was corrected.

- Page6, line 10: Again, how exactly is fragile landscape defined?
  Response: The "fragile landscape" was replaced by *"spatially highly variable landscape"*.

**References**

[revised manuscript text omitted]

---

## Author Response (AR2)

**Author's Response to Editor's Comments on *"Brief Communication: Evaluation and comparisons of permafrost map over Qinghai-Tibet Plateau based on inventory of in-situ evidence"**

Bin Cao[1,2], Tingjun Zhang[1], Qingbai Wu[3], Yu Sheng[3], Lin Zhao[4,5], and Defu Zou[5]

[1]Key Laboratory of Western China's Environmental Systems (Ministry of Education), College of Earth and Environmental Sciences, Lanzhou University, Lanzhou 730000, China
[2]Department of Geography & Environmental Studies, Carleton University, Ottawa K1S 5B6, Canada
[3]State Key Laboratory of Frozen Soil Engineering, Cold and Arid Regions Environmental and Engineering Research Institute, Chinese Academy of Sciences, Lanzhou 730000, China
[4]School of Geographical Sciences, Nanjing University of Information Science and Technology, Nanjing 210044, Chin
[5]Cryosphere Research Station on the Qinghai-Tibet Plateau, State Key Laboratory of Cryospheric Science, Cold and Arid Regions Environmental and Engineering Research Institute, Chinese Academy of Sciences, Lanzhou 730000, China

**Correspondence**: Tingjun Zhang(tjzhang@lzu.edu.cn)

**Response to Editor**

Dear Peter Morse,

Thank you for the constructive feedback, and the detailed assessment of the manuscript. Below we provide a point-to-point response to each comment, comments are given in black, responses are given in blue.

Bests,
Bin Cao

**Major:**

Thank you for your revisions. You have incorporated may of the reviewers' suggestions and the document has improved as a result. However, there are still two main issues that have to be resolved.

- First, this manuscript cannot be published as is, in part, because the language is poor. From the title to the last table cation there are numerous language-use problems that need to be addressed. Both of the reviewers noted this, and R1 suggested that you work with a native speaker to improve the writing. I agree. Please work with somebody who will help correct the mistakes and improve the clarity of the text. I have included several examples of how to simplify and clarify the text, but leave it to you to complete this task for the rest of the paper. Response: Thanks for the examples. The manuscript was carefully revised by native speaker, and all the changes are highlighted in the revised manuscript.

- Second, as written, the study is not repeatable. The analysis hinges on the methods used to simplify categorical data in to binary data (P4, Line 19), but these methods are not described. consequently, this also makes the results difficult to interpret and rely on. Response: We added a paragraph in Section 2.4 Statistics and evaluation of permafrost distribution maps (see below)

  *To conduct the map evaluations against measurements with binary information (presence or absence), it was necessary to develop classification aggregations for the existing maps. We argue that although the aggregation presented here simplifies the information available in these maps and may introduce uncertainty for further analyses, it is necessary in order to conduct inter-comparisons among them. For the IPA map, we consider the continuous and discontinuous permafrost zones to correspond to permafrost presence and the other zones*

*(sporadic permafrost, island permafrost, and non-permafrost) to correspond to permafrost absence by using the proportion of ground underlain by permafrost of 50% as a threshold. This is consistent with the threshold of the PZI map described below. For the $QTP_{TTOP}$ and $QTP_{Noah}$ maps, the permafrost distribution was derived using simulated mean annual ground temperature (thermally defined). In these maps, areas are classified into three type: permafrost, seasonally frozen ground, and unfrozen ground. Here, we merge the areas of seasonally frozen ground and unfrozen ground to yield areas of permafrost absence. For the PZI maps, specified thresholds are required for both the extent of permafrost region and permafrost area. Following Gruber (2012), only the areas with $PZI \geq 0.01$ were selected for further analysis, permafrost regions were defined as where $PZI \geq 0.1$, and permafrost area was calculated as PZI multiplied pixel area. A value of 0.5 was used as the threshold of permafrost presence and absence (Boeckli et al. , 2012; Azócar et al. , 2017).*

*There is a big mistake in the previous version. All the four permafrost types (continous, discontinous, sporadic, and island) in the IPA map were aggregated to permafrost presnece. This resulted in significant overestimatiton of permafrost distribution. Now we corrected the aggregation as described above.*

Once you have revised the manuscript, I will accept Reviewer 1's offer, and ask for a second review. I think that by then the manuscript will be ready for publication.

Thank you for your work, and I look forward to a revised version.

Best regards,

Peter

**Specific:**

Response: Thank you for the detailed editing. The language was corrected as suggested throughout the manuscirpt if not specified, and changes are highlighted in the marked-up manuscript version. We only listed the logic and technique comments below.

- P1, 12: Change "Cheng and Jin (2013)" to "Cheng and Jin, 2013", "Mu et al. (2017)" to "Mu et al., 2017", and "Wu et al. (2016)" to "Wu et al., 2016"
  Response: Corrected throughout the manuscript.

- P2, L2: In addition to what? The logic of this sentence do not follow from the previous.
  Response: The sentence was change to *"The QTP has also been included in ...."* to clarify.

- P2, L8: Gap in logic. Make it clear to the reader: How does having an improved evaluation improve the application of the existing maps?
  Response: This part was changed to

  *"A new inventory of this field evidence provides an opportunity to improve the evaluation of the existing permafrost maps. This is an important step in describing the current body of knowledge on permafrost mapping performance as well as identifying any possible bias. It is also critical for identifying priorities when updating these maps in the future. Additionally, an improved evaluation is a useful guide to selecting a map to use for permafrost and related studies, for example as a boundary condition for eco-hydrological model simulations."*

  to clarify.

- P2, L18: I think that Table A1 should be included within the main document. If the writing is improved in this section, it and the information on page 7, lines 20-24, can be incorporated quite well.
  Response: We agree. The table as well as the appendix A1 "Classification algorithm of in-situ permafrost presence or absence evidence" were merged into the main text (Section 2.1 Inventory of permafrost presence/absence evidence).

  *"In order to apply the permafrost presence or absence inventory more broadly, the degree of confidence in the data is estimated and provided in the inventory and in Table ??, although it is not used in this study. BH and SP provide direct evidence of permafrost presence or absence based on MGT and/or ground ice observations, and hence have high confidence (Cremonese et al., 2011). The data confidence derived from MAGST is classified*

*based on temperature and the length of the observation period. The evaluated GPR survey result was considered to have medium confidence."*

- P2, L19: MAGT is often used when referring to the temperature at the depth of zero annual amplitude. Do you simply mean mean ground temperatures?

- P2, L19: The way this sentence is written, it seems like temperature varies in length.
  Response: This is the response for above two specific comments. We changed the MAGT to "mean ground temperature (MAT)" throughout the manuscript to clarify. Yes, the borehole temperature used here veried depending on the depth of zero annual amplitude and measured depth. The sentence is changed to

  *"In this study, we used the mean ground temperature (MGT) measured from the borehole, the depth of which varies from meters to about 20 m depending on the depth of zero annual amplitude and borehole depth, to identify permafrost presence or absence."*

- P2, L21: Change "1 meter" to "1 m"
  Response: Corrected.

- P2, L21: This sentence doesn't make sense. Please clarify.
  Response: It is deleted.

- P2, L23: Define the thermal offset for readers. Lin et al., 2015, Permafrost and Periglacial Processes, discuss a reversed thermal offset in QTP. You may want to include something from it here.
  Response: Thermal offset is defined as *"the mean annual temperature at the top of permafrost minus MAGST"*, and reversed thermal offset is discussed. This GST part is changed to

  *"Thermal offset is defined as the mean annual temperature at the top of permafrost (TTOP) minus the mean annual ground surface temperature (MAGST) at a depth of 0.05 or 0.1 m. Although it is spatially variable depending on soil and temperature conditions, the magnitude of the thermal offset is small on the QTP compared with northern, high-latitude environments due to the prevalent coarse soil and low soil moisture content. The maximum thermal offset under natural conditions reported for the QTP is 0.79 °C (referenced as maximum thermal offset, $TO_{max}$) (Wu et al. , 2002; Wu et al., 2010; Lin et al, 2015). In this study, sites with MAGST + $TO_{max} \leqslant$ 0 °C are considered to be permafrost sites. The reversed thermal offset reported on the QTP was not considered here because thermal offset measurements are not available for all sites, and the influence of the reversed thermal offset is expected to be minimal due to its small magnitude (the value was reported as -0.07 °C by Lin et al (2015))"*

- P2, L31: What do you mean by clear? Can there be an opaque reflection? If the GPS survey is conducted in early summer, it could just be a measurement of the still frozen portion of seasonally frozen ground, rather than the active layer. This is why R#2 asked you to present criteria for the GPR surveys.
  Response: Cao et al. (2017) gave a detailed description for GPR data acquisition and processing, I copied the sentences below:

  - Section 3 (P4): *GPR profiles with unexpected attenuation were removed before further analysis.*
  - Section 3.1.1 (P4): *Measurements were conducted with a MALÅ ProEX GPR by using 200 MHz and 100 MHz unshielded antennas from late September to November in 2014.*
  - Section 3.1.1 (P5): *During late September to October, the thaw depth reaches its maximum, and thus, ALT can be obtained (Wang et al. , 2016). In November, the near-surface soils have frozen, and at deeper layer, there is still an unfrozen layer with high amount of unfrozen water owing to the so-called "zero curtain" effect (maintaining temperatures close to the freezing point over extended periods of time in freezing or thawing soils.*

This means (1) the opaque reflection was removed before further analysis, (2) GPR survey measured the active layer rather than seasonally frozen ground. We changed the sentence to

*"GPR data are from Cao et al. (2017), and were measured in 2014 between late September and November using 100 and 200 MHz antennas. The GPR survey depth is from about 0.8 m to nearly 5 m depending on the active layer thickness. The data were carefully processed by removing opaque reflections, and evaluated using direct measurements. The ability of GPR data to detect permafrost relies on the strong dielectric contrast between liquid water and ice (Moorman et al., 2003). Consequently, it is more difficult to discern the presence of permafrost in areas with low soil moisture content because it weakens this contrast (Cao et al., 2017). For this reason, the GPR data were only considered to indicate the presence of permafrost if an active layer thickness*

*could be established."*

to clarify.

- P3, L4: what is an arcsec? 3 arc second resolution?
  Response: Yes, it is the spatial resolution. the arcsec was changed to arc second(s) throughtout the manuscript. In this sentence, it is changed to
  *"...a DEM with a spatial resolution of 3 arc second"*

- P3, L8: Undefined "MAAT"
  Response: MAAT stands mean annual air temperature, and definition is added.

- P3, L18: neighbour?
  Response: Yes, it is changed to *"nearest-neighbour interpolation"* to clarify.

- P4, L4: add "PZI" before "map to allow..."
  Response: It is corrected as *"...into the PZI$_{global}$ map..."*

- P4, L19: As stated, this is totally unrepeatable. Clearly state how you decided to aggregate the categorical map data. e.g., how do you reclassify sporadic permafrost?
  Response: Please see our responses to the major comment 2.

- P5, L7: "fair" to "slight" and "slight" to "poor"
  Response: The sentenec is changed to
  *"...slight agreement for 0.2 $\kappa$ < 0:4, and poor agreement for $\kappa$ < 0:2."*

- P6, L6: "showed underestimated" ???
  Response: "showed" was deleted.

- P6, L9: Please consider dividing this paragraph into a set according to map type.
  Response: This paragraph is divided as two parts, the categorical and the PZI map.

*"Among the categorical maps, the QTP$_{TTOP}$ map achieved the best performance for permafrost distribution over the QTP with the highest $\kappa$ (0.58, moderate agreement) and PCC$_{tol}$ (82.8%), however, caution should be taken when interpolating the map. The QTP$_{TTOP}$ map was derived based on MODIS land surface temperature with temporal coverage of 2003–2012 (Zou et al., 2017). Though the MODIS land surface temperature time-series gaps caused mainly by clouds were filled using the Harmonic Analysis Time-Series (HANTS) algorithm (Prince et al., 1998), the surface conditions, especially vegetation and snow cover, were ignored. In this case, land surface temperature is underestimated in high or dense vegetation areas because it comes from the top of the vegetation canopy, and is overestimated in snow-covered areas where the cooling effects of snow are not considered. As a consequence, permafrost is likely overestimated in areas of high or dense vegetation and underestimated in regular snow covered areas. While the QTP$_{Noah}$ map performed slightly better (2.5 % higher) for permafrost area than the QTP$_{TTOP}$ map, it suffer from considerable underestimation of non-permafrost area (12.7% lower for PCC$_{NPF}$). Although the QTP$_{Noah}$ map was derived using a coupled land surface model (Noah), the poorer performance, especially for non-permafrost area (PCC$_{NPF}$ = 49.5%), is likely caused by the coarse-scale forcing dataset (0.1° resolution or $\sim$10 km) and by the uncertainty in the soil texture dataset (Chen et al., 2011; Yang et al., 2010). It is not surprising that the IPA map has slight agreement ($\kappa$ = 0.21) because fewer observations were compiled and the methods used were more suitable for high latitudes (Ran et al., 2012).*

*For the PZI map, the PZI$_{norm}$ and PZI$_{cold}$ maps were found to be in moderate agreement ($\kappa$ = 0.56 for the PZI$_{norm}$ map and 0.55 for the PZI$_{cold}$ map) with in-situ measurements, and performed slightly worse than the QTP$_{TTOP}$ map. The poor performance of the PZI$_{warm}$ map and underestimation of the PZI$_{norm}$ map indicated that permafrost over the QTP is more prevalent than most of the other regions even though the climate conditions, especially the MAAT, are similar. This is likely because of the high soil thermal conductivity due to coarse soil and the cooling effects of minimal snow (Zhang, 2005). Large differences of permafrost region (0.42 $\times$ 10$^6$ km$^2$, or 25% of the normal case) and area (0.49 $\times$ 10$^6$ km$^2$, or 49% of the normal case) were found for the three cases of the PZI$_{global}$ map, though the upper and lower bounds only changed about 5% for the PZI and $\pm$ 0.5 °C for the MAAT. The MAAT used in the PZI$_{global}$ map was statistically downscaled from reanalysis based on the lapse rate derived from NCEP upper-air (pressure level) temperatures. The land surface influences on surface air temperature, such as cold air pooling, were ignored (Cao et al. , 2017). This is important as winter inversions are excepted to be common due to the prevalent mountains over the QTP. In other words, permafrost may be underestimated in valleys due to the overestimated MAAT."*

**Tables and Figures**

**Table 1**

Not clear, poor writing: "Criteria of continuous means permafrost distribution is compiled as PZI range of [0.01–1]."
Response: It is changed to
*The continuous classification criteria means the permafrost spatial patterns is compiled or present as continuous value with a range of 0.01–1, e.g., permafrost zonation index in the PZI maps.*

"Some bias is expected for permafrost areas of $QTP_{TTOP}$ and $QTP_{Noah}$ as different QTP boundaries, lake and glacier data are used." is not really "note" material. move to discussion.
Response: It is moved to discussion (P6, L25–26)

**Table A1**

I think that Table A1 should be included within the main document. If the writing is improved in this section, it and the information on page 7, lines 20-24, can be incorporated quite well.
Response: It is moved to the main text as Table 1 in the revised manuscript.

**Figure 1**

Response: (1) The legend was moved in frame (a); (2) slope aspect unit (°) was added in frame (c), and the north direction was marked; (3) the ylab was changed to *"NDVI_max"*; (4) all axis titles now begin with a capital letter. The figure and caption were revised as below.

[revised manuscript text omitted]

BH = borehole temperature, SP = soil pit, GST = ground surface temperature, GPR = ground-penetrating radar, MGT = mean ground temperature, and MAGST = mean annual ground surface temperature. $TO_{max}$, the maximum thermal offset under natural conditions reported for the QTP, is 0.79 °C. Ambiguous means the data is not sufficient to determine permafrost conditions and is not included in the inventory.

**Table 2.** Summary and evaluation of existing permafrost maps over the Qinghai-Tibet Plateau

| Name | IPA | QTP$_{\text{TTOP}}$ | QTP$_{\text{Noah}}$ | PZI$_{\text{norm}}$ | PZI$_{\text{warm}}$ | PZI$_{\text{cold}}$ |
|---|---|---|---|---|---|---|
| Year | 1997 | 2017 | 2018 | 2012 | 2012 | 2012 |
| Method | – | semi-physical model | physical model | heuristic GLM | heuristic GLM | heuristic GLM |
| Classification  criteria | categorical | categorical | categorical | continuous | continuous | continuous |
| Scale | 1:10,000,000 | ~1 km | 0.1° (~10 km) | ~1 km | ~1 km | ~1 km |
| $PCC_{PF}$ [%] |  46.6 | 93.9 | 96.4 | 76.6 | 35.3 | 94.3 |
| $PCC_{NPF}$ [%] |  79.8 | 58.6 | 45.9 | 82.6 | 98.5 | 54.0 |
| $PCC_{tol}$ [%] |  57.0 | 82.8 | 80.7 | 78.5 | 55.1 | 81.7 |
| $\kappa$ |  0.21 | 0.58 | 0.52 | 0.56 | 0.36 | 0.55 |
| PF  region [$10^6$ km$^2$] | 1.63 | – | – | 1.68 | 1.42 | 1.84 |
| PF  area [$10^6$ km$^2$] | – | $1.06 \pm 0.09$ | 1.13 | 1.00 | 0.76 | 1.25 |
| Reference | Brown (1997) | Zou et al. (2017) | Wu et al. (2018) | Gruber (2012) | Gruber (2012) | Gruber (2012) |

Evaluations are conducted using 1475 in-situ measurements of permafrost presence or absence. GLM = generalized linear model, PF = permafrost. Norm (normal), warm, and cold means different cases and assumptions of parameters for permafrost distribution simulations in the PZI$_{\text{global}}$ map, details are from Table 1 of Gruber (2012). The continuous classification criteria means the permafrost spatial patterns is compiled or present as continuous value with a range of [0.01–1], e.g., permafrost zonation index in the PZI maps.

[Figure]

**Figure 1.** (a) The location of the QTP, and in-situ permafrost presence (PF) or absence (NPF) evidence distribution over the QTP, superimposed on the background of digital elevation model (DEM) with a spatial resolution of 30 arc second. (b) Number of field evidence located in NPF  and PF  regions.  SP means soil pit, GPR refers ground-penetrating radar, BH stands field evidence measured by borehole drilling,  and MAGST  is mean annual ground surface temperature. (c) Distribution of field evidence in terms of elevation (radius), slope (colored), and aspect (0/360° represents North).  Distributions of d) mean annual air temperature (MAAT), (e) scaled mean annual snow cover days (MASCD), and (f) annual maximum NDVI ($NDVI_{max}$) for field evidence (red line) comparing to the entire QTP (black line). Numbers in (d), (e), and (f) are mean values. Only the sites  with MAAT < 0 °C, which is the precondition for permafrost presence, were present in (d).

[Figure]

**Figure 2.** The permafrost classification results at in-situ evidence sites  shown on the (a) IPA, (b) QTP$_{TTOP}$, (c) QTP$_{Noah}$, (d) PZI$_{norm}$, (e) PZI$_{warm}$, and (f) PZI$_{cold}$ maps. The Cohen's kappa coefficient ($\kappa$ ), was derived from the  selected spatially highly variable landscapes (marked by black box) with 106 evidence sites. All the maps are re-sampled to the unprojected grid of SRTM30 DEM with a spatial resolution of 30  arc second ($\sim$1 km) to avoid maps bias  caused by different resolutions, geographic projection, and format. The boundary of QTP used in this study is marked by black line. Categorical classification is used for  the QTP$_{TTOP}$, QTP$_{Noah}$, and IPA maps, while continuous PZI was present for the PZI$_{norm}$, PZI$_{warm}$, PZI$_{cold}$ maps. The blank  parts in the PZI maps  are areas with PZI $< 0.01$.

~~Classification algorithm of in-situ permafrost presence or absence evidence from various methods Method Indicator Survey depth Permafrost Confidence degreeBH MAGT $\leqslant$ 0 °Cmeters to about 20 m presence highSP ground ice presence about 1.0–2.5 m presence highGST MAGST $\leqslant$ -2 °C & observations $\geqslant$ 3 0.05 or 0.1 m presence mediumMAGST $\leqslant$ -2 °C & observations $<$ 3 presence lowMAGST $>$ -2 °C & $MAGST + TO_{max} \leqslant 0$ °Cpresence lowMAGST $< 0$ °C & $MAGST + TO_{max} > 0$ °Cambiguous –MAGST $> 0$ °C absence mediumGPR clear permafrost reflection about 0.80–5.0 m presence medium~~

---

## Editor Decision (ED2)

**Author's Response to Editor's Comments on *"Brief Communication: Evaluation and comparisons of permafrost map over Qinghai-Tibet Plateau based on inventory of in-situ evidence"**

Bin Cao[1,2], Tingjun Zhang[1], Qingbai Wu[3], Yu Sheng[3], Lin Zhao[4,5], and Defu Zou[5]

[1]Key Laboratory of Western China's Environmental Systems (Ministry of Education), College of Earth and Environmental Sciences, Lanzhou University, Lanzhou 730000, China
[2]Department of Geography & Environmental Studies, Carleton University, Ottawa K1S 5B6, Canada
[3]State Key Laboratory of Frozen Soil Engineering, Cold and Arid Regions Environmental and Engineering Research Institute, Chinese Academy of Sciences, Lanzhou 730000, China
[4]School of Geographical Sciences, Nanjing University of Information Science and Technology, Nanjing 210044, Chin
[5]Cryosphere Research Station on the Qinghai-Tibet Plateau, State Key Laboratory of Cryospheric Science, Cold and Arid Regions Environmental and Engineering Research Institute, Chinese Academy of Sciences, Lanzhou 730000, China

**Correspondence**: Tingjun Zhang(tjzhang@lzu.edu.cn)

**Response to Editor**

Dear Dr. Cao,

Please make the revisions as suggested in the final review quoted here. I would like to read over the final version before recommending publication. Thanks again to you and your co-authors for your work.

Best regards,
Peter

Dear Peter Morse,

Thank you. Below we provide a point-to-point response to each comment, comments are given in black, responses are given in blue.

Bests,
Bin Cao

**Specific:**

Compared to the previous version, I think the authors have made a great effort to improve the clarity and language. They have adequately answered the questions the reviewers and editor had.
Given this, I suggest this revised version should be granted publication.
Yet, the authors might want to take into account the following remarks, of mainly technical details:

- Page 1, Line 3: '6' - > 'six'
  Response: Corrected.

- Page 6, Line 11-13: "Q25" and "Q75". Again, I know that means quantiles at different levels. However, that are not general acronyms for all communities. The authors should provide what the exact means of them are. I suggest you could spell out them, as they were used only here twice, e.g., "Q25" - > 25th percentile [or lower quartile].
  Response: Q25 is changed to 25th percentile, and Q75 is changed to 75th percentile.

- Page 8: Some data used have to be acknowledged, such as NDVI, GDEM2, etc.
  Response: We added

*"the GDEM2 dataset is downloaded from United States Geological Survey (`http://gdex.cr.usgs.gov/gdex/`), the NDVI datasets are derived and processed in the Google Earth Engine."*
in the Acknowledge.

- Page 10, Line 23-24: I know previous JGR papers only have article numbers. I am not sure "n/a-n/a" is allowed for TC publication standard.
  Response: "n/a-n/a' is removed.

- Page 11, Line 1-2: The DOI link seems not available. This paper may also need pages here. Please check them.
  Response: Yes, the DOI is not available though it is corrected. We deleted the DOI.

- Figure 2: It looks like missing lakes on the maps. Please check them again.
  Response: The lake inventory is added in Figure 2 (see below).

[revised manuscript text omitted]

---

## Author Response (AR4)

**Author's Response to Editor's Comments on *"Brief Communication: Evaluation and comparisons of permafrost map over Qinghai-Tibet Plateau based on inventory of in-situ evidence"**

Bin Cao[1,2], Tingjun Zhang[1], Qingbai Wu[3], Yu Sheng[3], Lin Zhao[4,5], and Defu Zou[5]

[1]Key Laboratory of Western China's Environmental Systems (Ministry of Education), College of Earth and Environmental Sciences, Lanzhou University, Lanzhou 730000, China
[2]Department of Geography & Environmental Studies, Carleton University, Ottawa K1S 5B6, Canada
[3]State Key Laboratory of Frozen Soil Engineering, Cold and Arid Regions Environmental and Engineering Research Institute, Chinese Academy of Sciences, Lanzhou 730000, China
[4]School of Geographical Sciences, Nanjing University of Information Science and Technology, Nanjing 210044, Chin
[5]Cryosphere Research Station on the Qinghai-Tibet Plateau, State Key Laboratory of Cryospheric Science, Cold and Arid Regions Environmental and Engineering Research Institute, Chinese Academy of Sciences, Lanzhou 730000, China

**Correspondence**: Tingjun Zhang(tjzhang@lzu.edu.cn)

**Response to Editor**

Comments to the Author:

Thank you again for your revisions. I have just a few additional minor revision that need to be treated before publication. These revisions are needed in order to clear up a few points. After I see these small changes, the manuscript will be ready to publish.

Please see my comments marked up in the attached "Comments to the Author"

Cheers,
Peter

Dear Peter Morse,

Thank you for the detailed editing for the manuscript. Below we provide a marked-up manuscript version showing the changes based on your comments.

Bests,
Bin Cao

[revised manuscript text omitted]

---

## Author Response (AR5)

**Author's Response to Editor's Comments on *"Brief Communication: Evaluation and comparisons of permafrost map over Qinghai-Tibet Plateau based on inventory of in-situ evidence"**

Bin Cao[1,2], Tingjun Zhang[1], Qingbai Wu[3], Yu Sheng[3], Lin Zhao[4,5], and Defu Zou[5]

[1]Key Laboratory of Western China's Environmental Systems (Ministry of Education), College of Earth and Environmental Sciences, Lanzhou University, Lanzhou 730000, China
[2]Department of Geography & Environmental Studies, Carleton University, Ottawa K1S 5B6, Canada
[3]State Key Laboratory of Frozen Soil Engineering, Cold and Arid Regions Environmental and Engineering Research Institute, Chinese Academy of Sciences, Lanzhou 730000, China
[4]School of Geographical Sciences, Nanjing University of Information Science and Technology, Nanjing 210044, Chin
[5]Cryosphere Research Station on the Qinghai-Tibet Plateau, State Key Laboratory of Cryospheric Science, Cold and Arid Regions Environmental and Engineering Research Institute, Chinese Academy of Sciences, Lanzhou 730000, China

**Correspondence**: Tingjun Zhang(tjzhang@lzu.edu.cn)

**Response to Editor**

Comments to the Author:
    Comments to the Author: Thank you for your revisions. Just a couple of technical edits to make now.

Cheers,
Peter

Dear Peter Morse,

    Thank you. Below we provide a marked-up manuscript version showing the changes based on your comments.

Bests,
Bin Cao

[revised manuscript text omitted]